# Biodegradable Poly(D-L-lactide-co-glycolide) (PLGA)-Infiltrated Bioactive Glass (CAR12N) Scaffolds Maintain Mesenchymal Stem Cell Chondrogenesis for Cartilage Tissue Engineering

**DOI:** 10.3390/cells11091577

**Published:** 2022-05-07

**Authors:** Clemens Gögele, Silvana Müller, Svetlana Belov, Andreas Pradel, Sven Wiltzsch, Armin Lenhart, Markus Hornfeck, Vera Kerling, Achim Rübling, Hannes Kühl, Kerstin Schäfer-Eckart, Bernd Minnich, Thomas Martin Weiger, Gundula Schulze-Tanzil

**Affiliations:** 1Institute of Anatomy and Cell Biology, Paracelsus Medical University, Nuremberg and Salzburg, 90419 Nuremberg, Germany; clemens.goegele@pmu.ac.at; 2Department of Biosciences and Medical Biology, Paris Lodron University Salzburg, 5020 Salzburg, Austria; thomas.weiger@plus.ac.at; 3Faculty of Materials Science, Technische Hochschule Nürnberg Georg Simon Ohm, 90489 Nuremberg, Germany; silvana.mueller@th-nuernberg.de (S.M.); belovsv61066@th-nuernberg.de (S.B.); andreas.pradel@heidolph.de (A.P.); sven.wiltzsch@th-nuernberg.de (S.W.); armin.lenhart@th-nuernberg.de (A.L.); markus.hornfeck@th-nuernberg.de (M.H.); vera.kerling@uni-bayreuth.de (V.K.); achim.ruebling@th-nuernberg.de (A.R.); hannes.kuehl@th-nuernberg.de (H.K.); 4Bone marrow Transplantation Unit, Medizinische Klinik 5, Klinikum Nürnberg, Paracelsus Medizinische Privatuniversität, 90419 Nuremberg, Germany; kerstin.schaefer-eckart@klinikum-nuernberg.de; 5Department of Environment & Biodiversity, Paris Lodron University Salzburg, 5020 Salzburg, Austria; bernd.minnich@plus.ac.at

**Keywords:** bioactive glass, CAR12N, cartilage regeneration, polymer infiltration, chondrogenesis

## Abstract

Regeneration of articular cartilage remains challenging. The aim of this study was to increase the stability of pure bioactive glass (BG) scaffolds by means of solvent phase polymer infiltration and to maintain cell adherence on the glass struts. Therefore, BG scaffolds either pure or enhanced with three different amounts of poly(D-L-lactide-co-glycolide) (PLGA) were characterized in detail. Scaffolds were seeded with primary porcine articular chondrocytes (pACs) and human mesenchymal stem cells (hMSCs) in a dynamic long-term culture (35 days). Light microscopy evaluations showed that PLGA was detectable in every region of the scaffold. Porosity was greater than 70%. The biomechanical stability was increased by polymer infiltration. PLGA infiltration did not result in a decrease in viability of both cell types, but increased DNA and sulfated glycosaminoglycan (sGAG) contents of hMSCs-colonized scaffolds. Successful chondrogenesis of hMSC-colonized scaffolds was demonstrated by immunocytochemical staining of collagen type II, cartilage proteoglycans and the transcription factor SOX9. PLGA-infiltrated scaffolds showed a higher relative expression of cartilage related genes not only of pAC-, but also of hMSC-colonized scaffolds in comparison to the pure BG. Based on the novel data, our recommendation is BG scaffolds with single infiltrated PLGA for cartilage tissue engineering.

## 1. Introduction

Articular cartilage covers the surface of joints, allows smooth sliding motions between two joint bodies, and provides uniform pressure distribution on the subchondral bone in response to joint loading through its unique viscoelasticity [1]. This is primarily accomplished by the highly specific and exquisitely organized cartilage extracellular matrix (ECM), formed by a few (1.4 × 10^4^ cells/ mm^3^) specialized cartilage cells (chondrocytes) [2,3]. The major components in hyaline joint cartilage include collagen types II, VI, IX, XI, sulphated glycosaminoglycans (sGAG), the water-binding cartilage proteoglycan (PG) aggrecan and structural proteins such as cartilage oligomeric matrix protein (COMP) and matrilin 1 and 3 [1]. Due to the facts that the chondrocytes are only supplied by diffusion from the synovial fluid with no direct blood access, hyaline cartilage is bradytrophic and lacks blood or lymphatic vessels and nerves [2,4,5]. One can assume that this also greatly reduces the ability of cartilage to regenerate since it is vulnerable to multiple harmful factors such as trauma, overloading, aging and inflammation [6]. One of the most common late sequelae of cartilage injuries is the chronic joint disease posttraumatic osteoarthritis (OA), for which there are still no cures that could interfere with the etiopathogenesis of OA and thus alter its natural course [7,8,9]. For this reason, autologous chondrocyte implantation (ACI) or matrix-associated chondrocyte implantation (MACI) are used to counteract the progressing inflammatory response induced by cartilage defects [10,11,12,13,14,15]. The MACI is a newer technique in which chondrocytes were colonized in a three-dimensional (3D) arrangement and the matrix serves as a cell carrier, protecting them from mechanical overloading [16]. Despite the very encouraging results obtained so far, also in the long-term studies [17,18,19,20], some limitations remain, such as the extensive costs particularly for the cell preparation [21], two surgical procedures required [22] and the loss of the specific cellular phenotype during expansion in vitro [23,24]. Senescence [25] and dedifferentiation [23,24] of cells must be prevented. When mesenchymal stem cells (derived from the bone marrow (B), adipose tissue (A), synovium (S), or periosteum (P)) are used, successful chondrogenic differentiation should occur and senescence [26] as well as hypertrophy be prevented [27,28,29]. For this reason, materials should be used for cartilage tissue engineering that provide incentives for cells to adhere, survive, proliferate, and build up a new and specific mechano-competent ECM [30,31,32]. Such stimuli are, on the one hand the geometric structure (three dimensional [3D] topology) and thus, the corresponding interconnective porosity and permeability of the scaffolds to enable cell immigration and nutrition [30,33] and, on the other hand, the release of bioactive molecules (e.g., growth factors, peptides and essential amino acids) [34] or ions [35]. Bioactive glasses (BG) were developed for this reason, to cause a release of different ions over time stimulating the cells. While the first bioactive glasses described by L. Hench were developed primarily for bone regeneration, today there are a variety of bioactive glasses with different ion compositions available that can also be used for other tissue applications including nerve, skin and cartilage regeneration [36,37,38,39]. The advantage of BG is not only the temporal degradation rate in vitro or in vivo but also the easy processability, which depends on the ion composition [40]. 

However, a decisive disadvantage of BG compared to many other materials is that they can be very brittle [41]. The brittleness does not only depend on the ion composition, but also on the topology, e.g., distribution of the pores, the strut thickness in the case of scaffolds and the overall porosity. In order to meet both, the cellular requirements by means of bioactivity and porosity on the one hand and to withstand the biomechanical conditions in the body on the other hand, so-called composite materials were developed [41,42]. Composites of BG and diverse synthetic polymers (polycaprolactone (PCL) [43], poly(ethylene glycol) (PEG) [44], poly(glycolic acid) (PGE) [45], polylactid acid (PLA) [46], poly (D-L-lactide-co-glycolide) (PLGA) [47] Poly(L-lactide) (PLLA) [48] and many more) have been proven to be suitable as bioactive carriers for different cell types due to their good biomechanical properties, high biocompatibility, non-toxicity and biological integrity. 

In this study, a composite cartilage scaffold was developed by combining a newly patented BG (CAR12N) [49] with PLGA utilizing a novel and simple solvent phase infiltration based approach to integrate a sponge of PLGA into a 100% BG scaffold by maintaining high porosity. The objective of this work was to evaluate the distribution of the polymer, and the associated change in porosity. Cytotoxicity of the scaffolds was assessed in order to subsequently achieve a long-term culture of up to 35 days with articular chondrocytes and mesenchymal stem cells (MSCs) as chondrogenic cells. Quantitative measurements of DNA and metabolic activity allowed conclusions to be drawn about the maintenance of proliferation, while the synthesis of sGAGs and expression of cartilage related extracellular matrix proteins provided ECM formation of the cells. Furthermore, the progress of chondrogenesis in the newly developed composite scaffold was evaluated by relative gene expression of the most prominent cartilage-specific markers (Figure 1).

## 2. Materials and Methods

### 2.1. Bioactive Glass (BG) Scaffold Fabrication (CAR12N)

The BG scaffolds (CAR12N) were produced by our recently patented glass composition (DE 10 2018 114 946 B3, 2019) (see Table 1). The detailed BG scaffold preparation was previously described [49]. After the sintering process [360 °C/hour (h) until 380 °C with no holding time, 46 °C/h until 520 °C with no holding time and 360 °C/h until 655 °C with 40 min (min) holding time] the scaffolds were stored in a desiccator for dry storage until further use. Before using the scaffolds in the cell culture (for colonization, for cytotoxicity assay see description below) the scaffolds were leached out (0.1 g scaffold mass on 1 mL liquid) for 8 days in a 0.1 M hydrochloric acid (HCl), stored for a minimum of 8 days and a maximum of 28 days in pyrogenic Aqua distilled (A. dist.; Carl Roth GmbH, Karlsruhe, Germany) and preincubated for two days in culture medium.

### 2.2. Preparation of PLGA-Infiltrated CAR12N Scaffolds

The scaffold systems were prepared according to our patent DE 10 2019 124 879 A1. Poly(D,-L-lactide-co-glycolide) (PLGA; 85:15; RG 858S; Evonik, Essen, Germany) was used for the scaffold infiltration. PLGA was dissolved with dioxane (Carl Roth GmbH) in a 5:95 relation. The following calculation was used to determine the dioxane quantity required: $M_{dioxane}=\frac{M_{PLLA}}{5}\times 95$. The PLGA was completely dissolved (30 min at 75 °C) to achieve a homogeneous infiltration of the BG scaffold. The PLGA solution infiltrated the pure BG scaffolds either one-time (1P), twice (2P) or three times (3P). Infiltration was performed by a drop technique. The first drop on the top, the second drop was added on the bottom and the third drop was added again on the top region. Each drop contained in average 0.0028 g of PLGA. The coated scaffolds were then immersed in a 60 °C A. dist. for 1–2 min for the PLGA to precipitate. The infiltrated scaffolds were dried for 30 min at 100 °C and then stored in the desiccator until further use.

### 2.3. Scaffold Characterization by Light Microscopy

To better distinguish the PLGA from pure glass, the PLGA solution (see description above) was dyed with the dye Sudan IV (scarlet red, Carl Roth GmbH). The dried scaffolds stained with Sudan red PLGA (BG as control, 1P, 2P, 3P) were then vertically cut in half with a razor blade. Light microscopic (Olympus SZH10 Research Stereo, Hamburg, Germany) images were taken with the camera system (Zeiss, Axio Cam ERc5s/Olympus Highlight 3001, Wetzlar, Germany) depicting the upper third (top region), middle third (middle region), and lower third (bottom region) of the PLGA-infiltrated scaffolds to evaluate the distribution of PLGA. Three images of three scaffolds from three different batches were taken and the percentage of PLGA, glass, and pores was calculated using GNU Image Manipulation Program (GIMP) (free software version 2.10.24) and ImageJ version 1.53i (National Institutes of Health, MD, USA). 

### 2.4. Pore Perimeter, Strut Length Measurement and Porosity

The pore perimeter was calculated from the native scaffold pictures with the “Freehand selections tool” from the ImageJ software. The strut length and diameter were calculated from the native scaffold pictures with the “straight line tool” from the ImageJ software. Porosity was calculated with the following formula: $\Phi =\left(1-\frac{\rho }{\rho_{0}}\right)\times 100\%$ The density *ρ* of the unsintered bioactive glass in the form of a fragment after the casting process and the density *ρ*_0_ of a scaffold were used for the calculation. The theoretical density of the glass is determined using the BatchMaker program (ilis GmbH, Erlangen, Germany). The exact chemical composition (Table 1) had to be entered to calculate the properties of the glass. For the calculation of the real density, a bubble-free piece of glass was used. The density of the scaffolds was determined by the mass of the samples. The porosity refers only to the open pores and not to the pores inside the struts.

### 2.5. pH Measurements

A pH meter (PCE-PH20S, PCE Americas Inc., Jupiter, FL, USA) was used to measure the pH values. The pH value was determined every day during the leaching process of the scaffold variants in 0.1 M HCl at room temperature (RT, 22.02 ± 0.11 °C), once a week (over 4 weeks) during scaffold storage in A. dist. (at RT) and every day during incubation in growth medium (37 °C). Four different BG batches were tested in this study. 

### 2.6. Mechanical Strength Testing

This test was used to compare the compressive strength and, more importantly, the fracture behavior of unleached and leached scaffolds, either PLGA-infiltrated or not. The compressive strength was the resistance of the scaffold to the action of pressure and could be calculated as the quotient of the failure load and the cross-sectional area. The tests were carried out with the GT200 particle compression tester (M-TECH Automation GmbH, Cologne, Germany). Measurements were shown as Appendix A.

### 2.7. Cell Isolation, Cultivation and Colonization

The pACs were isolated from 4 healthy knee joints from 4–6 months old female pigs derived from the abattoir. The harvested cartilage pieces were cut into small slices (2 mm) and digested by collagenase NB5 derived from *Clostridium histolyticum* (1 mg/mL, Nordmark, Uetersen, Germany) diluted in chondrocyte growth medium (96% [*v*/*v*] DMEM/Ham’s F-12 [1:1] with stable L-glutamin, 1% [*v*/*v*] amphotericin B, 1% [*v*/*v*] MEM amino acids, 1% [*v*/*v*] penicillin/streptomycin, 1% [*v*/*v*] ascorbic acid supplemented (all products from Carl Roth GmbH) with 10% fetal calf serum (FCS) (Pan-Biotech GmbH, Aidenbach, Germany) overnight in a rotatory device at 37 °C. After sieving (100 µm cell sieve, TPP, Trasadingen, Switzerland) and washing, isolated chondrocytes were cultured in chondrocyte growth medium in T175 flask (CellPlus, Sarstedt AG, Nümbrecht, Germany). For expansion, chondrocytes were detached with 0.05%/0.02% trypsin/ Ethylenediaminetetraacetic acid (EDTA) solution (Carl Roth GmbH) and seeded in T175 flasks. Every second day growth medium was changed. Voluntary male bone marrow donors (average age of 29 ± 12.1) were used to isolate their hMSCs. The ethical commission of the Bavarian medical association (no.17074, 6 February 2018) approved the harvesting and usage of human derived tissues for experiments. The experiments were performed in accordance with the Declaration of Helsinki. The bone marrow blood transplantation unit of the Nuremberg General Hospital took the biopsies from the iliac crest. MSC growth medium was mingled with fresh bone marrow blood (4:1) and transferred into a T175 flask (Sarstedt). The components of hMSC’s growth medium were: Dulbecco’s modified Eagle’s medium (DMEM with stable glutamine [3.7 g/L NaHCO_3_ and 4.5 g/L D-glucose], Pan-Biotech GmbH), 5% human growth factor-rich human Platelet Lysate (PL) solution (PL BioScience, Aachen, Germany), 0.04% heparin (PL BioScience), 1% [*v*/*v*] amphotericin B, 1% natrium pyruvate (Pan-Biotech GmbH), 1% penicillin/streptomycin (Carl Roth GmbH). Fresh medium (10 mL) was added to each flask after 3 days. A complete medium removal was carried out on day 6. Two washing steps of the adherent cells were performed with phosphate-buffered saline (PBS, Carl Roth GmbH) and an incubation in 20 mL fresh MSC growth medium was carried out until a confluency of 80–90% was reached. The cell isolation and cultivation protocol used is described in our last publication [49]. 

Cubic scaffolds (6.64 ± 0.40 mm length; 6.64 ± 0.40 mm width; 3.32 ± 0.45 mm high) without or with PLGA were leached out in 0.1 M HCl solution for eight days after production. For a minimum of eight days the scaffolds were stored in A. dist. with very low pyrogen content before they were incubated for two days in culture medium. Then, the scaffolds were transferred to a TubeSpin bioreactor tube (TPP) and colonized with 5 × 10^5^ primary cells pACs or hMSCs (in passages between 3 and 4) per scaffold for dynamic seeding in 5 mL chondrocyte or MSC growth medium. A homogenous cell distribution on the scaffold was reached by using a rotatory device (Bartelt GmbH, Graz, Austria) with 36 rpm at 37 °C. Every second day medium changes were executed until the cultivation was stopped at day 1, 3, 7, 21, and 35. Three independent experiments were performed with cells isolated from three different donors.

### 2.8. Cytotoxicity Testing

The murine fibroblast cell line L929 (derived from subcutaneous tissue, strain C3H/An, from Cell Line Services, Eppelheim, Germany) and synovial fibroblast cell line K4IM (derived from a male human synovium; with SV40 T antigen immortalized fibroblast-like synoviocytes, cell line collection C616) were used for biological evaluation of cytotoxicity according to the international standard DIN EN ISO 10993-5 2009-10 norm. With an initial cell density of 1.0 × 10^4^ cells/cm^2^, thawed primary pACs and hMSCs (three different donors) and L929 fibroblasts (cultivated in the same cultivation medium as chondrocytes, the components are listed above) were cultured in growth medium until 80–90% confluence at 37 °C and 5% CO_2_ was reached. A change of growth medium was carried out three times a week. Determination of potential cytotoxic effects was evaluated with extracts from three different scaffold batches. Cell-specific growth medium was used as extraction medium. Ten scaffolds (0.1769 g) of each scaffold variant (BG, 1P, 2P, 3P) were incubated in 1.769 mL extraction medium at 37 °C and 5% CO_2_ for 48 h under aseptic conditions using sterile, chemically inert cell culture plates (Sarstedt). The ratio scaffold/ extraction medium was 100 mg scaffold in 1 mL medium. Cells (pACs, hMSCs, L929) were seeded in 96-well cell plates (Sarstedt) with an initial density of 1.0 x 10^4^ cells/cm^2^ and the cells were incubated for 24 h at 37 °C and 5% CO_2_ to allow cell adherence. After a removal of growth medium, cell cultivation in 100 µL extraction medium or control solutions per well was performed for 24 h 37 °C and 5% CO_2_. As a positive control, a 10% dimethyl sulfoxide solution (DMSO, Carl Roth GmbH) in cell type specific growth medium was used. As a negative control, cell type specific pure growth medium was used. In the Appendix A, the test was extended by using 48 h incubated medium and 10% DMSO in 48 h incubated medium, to evaluate the influence of loss of higher temperature sensitive components of the growth medium. A complete removal of the supernatant after 24 h of incubation with the respective extracts as carried out. To each well a mix of 80 µL of the respective growth medium and 20 µL [3-(4,5-dimethylthiazol-2-yl)-5-(3-carboxymethoxyphenyl)-2-(4-sulfophenyl)-2H-tetrazolium, inner salt; MTS] solution (CellTiter 96^®^Aqueous One Solution Cell Proliferation Assay, Promega GmbH, Walldorf, Germany) was added. The absorbance was measured photometrically at a wavelength of 490 nm (Tecan Austria GmbH, Grödig, Austria) after additional 2 h incubation of the cells under standard culture conditions. The test was performed with scaffold cultures seeded with three different pAC or MSC donors. Three different scaffold batches were tested with each donor. The protocol used is described in detail in our previous publication [49]. 

### 2.9. Viability Assay

The viability assay was performed to analyze if the cells remained alive or died. The live/dead staining solution consisted of 1 µL propidium iodide (PI, 1% stock solution in A. dist., ThermoFisher Scientific, Darmstadt, Germany) and 5 µL fluorescein diacetate (FDA, stock solution: 3 mg mL-1 in acetone, Sigma-Aldrich, St. Louis, MO, USA) in 1 mL PBS. Cell viability on the scaffolds was determined after 7, 21 and 35 days of cultivation. For performing the live/dead staining, the scaffolds were removed from growth medium, transferred to a microscopic cover slide and 50 µL of stain solution was added. After a 5 min incubation period at RT, the fluorescence of live (green) and dead (red) cells was monitored using a Leica SPEII DMi8 confocal laser scanning microscope (CLSM, Leica, Wetzlar, Germany). The protocol used followed the procedure described in our previous publication [49].

### 2.10. Calculation of the Viability, Colonized Scaffold Surface and Cell Seeding Efficiency

Based on the viability assay, three different pictures with vital and dead cells from three independent experiments were evaluated. The pictures were “split” with the LAS X software (Leica, 3D program) in the green and red channels. The amount of vital (green) and dead (red) cells on the scaffold surface was calculated with the “analyzer” feature in the program. For the calculation of the viability per scaffold surface, the amount of vital (green) and dead (red) cells were evaluated. The quantification of viability was calculated as follows: $\mathrm{viability}\;\left(\%\right)=\frac{amount\;of\;live\;cells}{amount\;of\;live\;cells+dead\;cells}\times 100$ For determining the “colonized scaffold surface” only the green channel was taken and the area of vital cells was chosen. The protocol applied has been described previously in detail [49]. The cell seeding efficiency was calculated based on the number of unattached cells that ended up in the residual medium according to Dozza et al. [50] with the following calculation:$$cell\;seeding\;efficiency\;\left(\%\right)=1-\frac{number\;of\;unattached\;cells\;}{number\;of\;cells\;initially\;seeded}\times 100$$

Cell counting was performed with a hemocytometer after 1 and 3 days. 

### 2.11. Scanning Electron Microscopy

After a fixation of the scaffolds in 2% paraformaldehyde (PFA, Morphisto GmbH, Frankfurt am Main, Germany), 2.5% glutaraldehyde (Carl Roth GmbH) in PBS without Ca^2+^ and Mg^2+^ overnight at 4 °C, the scaffolds were washed in PBS four times for 15 min and contrasted with 1% osmium tetroxide (OsO_4_, Carl Roth GmbH). Then, gentle rinsing of the scaffolds four times, each 15 min in PBS was performed before the dehydration in an ascending ETOH series (70%, 80%, 90% and 96% ETOH, each 30 min) and three times in 100% ETOH for 15 min each started. Followed by a critical point drying, mounting onto specimen stubs and sputtering (agar auto carbon coater, BalTEC COD 030, Agar scientific Ltd., Essex, UK) with a thin layer of carbon (≈13 nm) of the scaffolds. SEM images were taken under an ESEM XL30 (FEI Europe B.V., Eindhoven, The Netherlands) at an accelerating voltage of 25 kV. The described protocol follows the description in our previous publication [49].

### 2.12. Measurement of Total DNA and sGAG Amount

By CyQUANT^®^ NF Cell Proliferation Assay the influence of respective treatment on cell division was examined after 1, 3, 7, 21, and 35 days. A serial dilution of calf thymus DNA stock solution (1 mg mL-1, Sigma Aldrich) with TE-buffer (TRIS EDTA buffer, 10 mM TRIS (pH 8.0), 1 mM EDTA in A. dist.) serves a standard curve. A mix of 25 µL of the serial calf thymus DNA dilution with 25 µL of CyQuant dye solution (1× Hank’s Balanced Salt Solution (HBSS, Carl Roth) + dye binding solution 1:250 (ThermoFisher Scientific Inc., Waltham, MA, USA)) were used for the standard curve. The scaffolds, from time point 1, 3, 7, 21 and 35 days were transferred to RNase and DNase free 2 mL safe seal tubes (Sarstedt) with 50 µL of the proteinase K digestion buffer (50 mM TRIS/HCl, 1 mM EDTA, 0.5% Tween 20) containing 0.5 mg mL-1 proteinase K (PanReac, ApplyChem, Darmstadt, Germany). Seven millimeter stainless-steel beads (RNase- and DNase-free, sterile, Qiagen, Hilden, Germany) were used to homogenize the samples with the Tissue Lyser (Qiagen, 50 Hz, 5 min, RT). A 250 µL proteinase K digestion buffer containing 0.5 mg mL-1 proteinase K was added. The sample digestion was performed for 16 h at 56 °C under continuous shaking. By freezing the samples at −20 °C for halve an hour the enzymatic reaction was stopped. The thawed samples were centrifuged for 15 min at 16483.55 g. A mix of 10 µL of each sample with 150 µL TE buffer was made before 25 µL of the sample dilution were transferred in triplicate into a black, flat-bottom, 96-well plate (Brand GmbH, Wertheim, Germany). A total of 25 µL of the dye solution (1× HBSS + dye solution 1:250) was mixed with the samples. Coved plates were incubated at 37 °C for 1 h. The fluorescence measurement of each well was performed in triplicate at 485 excitation/530 emission nm in a fluorometric plate reader (Tecan Austria GmbH). Three independent experiments with cells derived from three different donors were performed. The procedure described has been published previously [49].

The DMMB Assay was performed with the same supernatant as in the CyQuant Assay. Samples were adequately diluted and, the DMMB solution (ApplyChem) was added (40 mM glycine (Sigma-Aldrich), 40 mM NaCl (Carl Roth GmbH) at pH 3 and DMMB [8.9 mM in ethanol (ETOH)]). For the standard curve chondroitin sulfate (Sigma-Aldrich) was taken. By using a genios spectral photometer (Tecan Austria GmbH), the absorption shift was measured at a wavelength of λ = 633 nm to λ = 552 nm. Three independent experiments with cells derived from three different donors were performed. The data were presented as GAG per DNA amount. 

### 2.13. Metabolic Activity Evaluation

The metabolic activity of the cells on scaffolds was determined with the CellTiter-Blue^®^ Cell Viability Assay (Promega GmbH) after the time points of 7 and 35 days. Scaffolds were incubated with a mixture of 100 µL of the culture medium and 25 µL of CellTiter Blue solution. After 2 h of incubation, 100 µL liquid were transferred into a fresh and new well and were than measured. The absorbance of each sample was measured in triplicates at 570 nm and 600 nm in a fluorometric plate reader (Tecan Austria GmbH). Reference values were the uncolonized scaffolds and were calculated as 0% metabolic active.

### 2.14. Immunocytochemical Staining

The immunocytochemical stained proteins were visualized by the CLSM. After the cultivation time (21 days) scaffolds were fixed with 4% PFA and after a washing step with TRIS buffered saline (TBS: 0.05 M TRIS, 0.015 M NaCl, pH 7.6) the scaffolds were incubated with protease-free donkey serum (5% diluted in TBS with 0.1% Triton X 100 for cell permeabilization) for 20 min at RT. Incubation was carried out overnight at 4 °C with primary antibody (Table 2: collagen type II, cartilage proteoglycans, SOX9). Then, the samples were washed with and incubated with donkey-anti-rabbit-Alexa 488 (Invitrogen, Carlsbad, CA, USA) or donkey-anti-mouse-cyanine-3 (Cy3, Invitrogen) coupled secondary antibodies (diluted 1:200 in TBS, see Table 2) for 1 h at RT. In total, 10 μg/mL 4′,6′-diamidino-2-phenylindol (DAPI, Roche, Mannheim, Germany) was used to counterstain cell nuclei. After the washing steps (3×) of the immunolabeled scaffolds with TBS, the examination was performed with the CLSM. Three independent experiments were performed with representative microscopic fields. For the calculation of the “collagen type II, cartilage proteoglycan and SOX9 area per cell”, three pictures of three independent experiments were taken and split with the 3D CLSM Leica software in the “green”, “red” and “blue” channels. The green and red area (%) was measured and related to the amount to blue dots (=cell nuclei).

### 2.15. RNA Isolation

After 7 and 21 days, BG and 1P scaffolds colonized with pACs and hMSCs were snap-frozen (each *n* = 3). Homogenization of the scaffolds and native porcine articular cartilage (*n* = 3) were performed in RLT-buffer (Qiagen) with the Tissue Lyser for 5 min at 50 Hz. According to the manufacturer’s instructions (Qiagen) RNA isolation and purification was carried out with RNeasy Mini kit, including on-column DNAse treatment. Nanodrop ND-1000 spectrophotometer (Peqlab, Biotechnologie GmbH, Erlangen, Germany) was used to monitor (260/280 absorbance ration) the purity and quality of the RNA samples. The procedure described is consistent with the protocol outlined in our previous publication [49].

### 2.16. Quantitative Real Time PCR

The QuantiTect Reverse Transcription Kit was used for reverse transcription of 120 ng of total RNA for cDNA synthesis. For each semiquantitative real-time PCR (qRT-PCR) reaction 35 ng cDNA were used for TaqMan Gene Expression Assays (Life Technologies, Carlsbad, CA, USA) with primer pairs for aggrecan (ACAN), collagen type I (Col1A1), collagen type II (Col2A1), collagen type IX (Col9A3), collagen type X (Col10A1), collagen type XI (Col11A1), cartilage oligomeric matrix protein (COMP), forkhead boxO 1 (FOXO1), SRY (sex-determining region Y)-box 9 protein (SOX9) and the β-actin (ACTB) as the reference gene (Table 3) real time PCR detector StepOnePlus (Applied Bioscience (ABI), Foster City, CA, USA) thermocycler with the program StepOnePlus software 2.3 (ABI) was used for qRT-PCR performance. The relative expression of the target gene by the cells on the scaffolds was normalized to the ACTB expression and calculated for each sample using the ΔΔCT method as described by Schefe et al. 2006 [51].

### 2.17. Statistical Analysis

The statistical description refers to all experiments performed. Data from all experiments are expressed as mean values with standard deviation (SD). GraphPad Prism8 (GraphPad software Inc., San Diego, CA, USA) was used to perform statistics. A ROUT outlier test (Q = 1%) was performed. Shapiro–Wilk test was used for the analysis of the normal distribution of the results. A two-tailed one-way ANOVA (Fisher) was used followed by Tukey´s multiple comparison post hoc testing to compare groups. The level significance (CI) was set at *p* values of ≤0.05 (*) and *p* values of ≤0.0001(****). Power of tests was 0.8. Four different scaffold batches with three different scaffolds were taken for the evaluation of the PLGA, glass and pore distribution, the glass strut length, the glass strut diameter, the pore perimeter and the porosity. Three different scaffold batches were taken for the stiffness measurement. Four different batches were taken for the pH measurement and three different batches were taken for the measurement of the scaffold parameters (two-way ANOVA was used by Tukey´s multiple comparison post hoc testing). Independent experiments (3–4) with cells from three to four different donors were included.

## 3. Results

### 3.1. PLGA Distribution within the Scaffolds

The leached scaffolds were vertically cut in half with a scalpel to evaluate the morphology in the top, middle and bottom regions of the scaffold. The red stained PLGA was detectable in every scaffold region but not homogenously distributed inside the scaffolds (1P, 2P, 3P). Due to using the “solvent phase infiltration” technique to introduce the PLGA into the scaffolds, a higher amount of PLGA accumulated in the top region (Figure 2). The evaluation of the percentage distribution of PLGA, glass and pores per picture gave an overview of their position within the scaffold and therefore, the scaffold characteristics. The PLGA amount was higher in the top and bottom regions of the 1P, 2P and 3P scaffolds in comparison to the middle region. The PLGA amount was significantly higher in the bottom region in the 2P scaffolds in comparison to the 1P bottom region and the middle region of the 2P scaffolds (Figure 2M). The average glass content was over 55%. Nevertheless, there was also a large scatter within the scaffold regions. In the top region of the 3P scaffold, there was significantly lesser glass content than in the top region of the 1P scaffold (Figure 2N). The pore fraction per scaffold region was also uniform within a variant. The pure BG scaffolds depicted the highest pore fraction (Figure 2O). It must be mentioned that these calculations refer only to light microscopic images of scaffold sections.

### 3.2. Scaffold Topology and Polymer Infiltration

Based on the light microscopic images, the glass strut length, glass strut diameter and pore perimeter were measured. The glass strut lengths were generally greater than 150 µm. The glass struts were significantly longer in the bottom region compared to the top regions of all four scaffold variants. Additionally, there were significant differences between the BG and 3P, the 2P and the 3P scaffolds within the bottom regions. The strut length was in top region of the BG scaffolds significant smaller than in comparison to the 2P and 3P scaffolds (Figure 3A). In contrast to the glass strut length, the glass strut diameter in the middle region was the highest in every scaffold variant in comparison to the top and the bottom regions (significant differences in the BG and the 1P scaffolds). On average, the glass strut diameter was larger than 50 µm for all four scaffold variants. Significant differences of the strut diameter could be also calculated between the different variants in the top region, for example BG and 3P; 1P and 3P; BG and 2P (Figure 3B). In general, the pore perimeter in the top, middle, and bottom regions of all four scaffold variants was greater than 400 µm. In tendency, the pore perimeter in the bottom region of all four scaffold variants was larger than in the top and middle regions. The bottom region had a significantly higher perimeter in the 1P and the 3P scaffolds in comparison to the top or the middle regions. Indeed, the perimeter of the 3P scaffolds in the top region is significantly smaller than in the BG, the 1P and the 2P scaffolds (Figure 3C). Porosity slightly decreased with increasing PLGA infiltration (not significantly), with the pure BG scaffold having a porosity of over 86% (86.10 ± 2.98), while the 3P scaffold had a porosity of less than 80% (78.14 ± 2.02) (Figure 3D).

### 3.3. pH Profile during Leaching Process and Gel Layer Formation

The alkali metal oxide content was reduced by selective leaching via an acid surface treatment. Alkali metal oxides, such as Na_2_O and K_2_O, were added to the CAR12N to act as network converters and thus, served to adjust the melting and sintering temperature. Through the selective leaching process, Na^+^ and K^+^ ions were dissolved from the glass matrix by HCl. This resulted in an ion exchange between Na^+^/K^+^ in the network and H^+^ in the acidic solution. This led to a leached gel layer with minimal alkali content. It was observed that for all four scaffold variants (BG, 1P, 2P and 3P) there was a continuous but not significant increase in pH over a period of 8 days. After one day most of the alkalis went into solution and after 8 days almost none (Figure 4A). In order not to exceed the leaching process, the scaffolds were subsequently transferred to A. dist. The leaching process has a duration of 8 days. Over the time course of up to 28 days, it could be shown for the BG scaffolds that the pH displayed an increasing phase and had already reached its plateau after 10 days. The 1P, 2P and 3P scaffolds showed a similar pH course as the BG scaffolds. For the 1P scaffold, there was a significant pH increase from day 1 to day 7. In addition, the pH value of A. dist. was measured to obtain a reference value. As indicated by the manufacturer (pyrogen-free sterile water, Carl Roth GmbH), this value remained constant at 7.3 over the measured period of 28 days. Compared to the BG scaffolds, the pH value of A. dist. was significantly lower on days 7, 14, 21 and 28 (Figure 4B). To provide evidence that the scaffolds no longer caused a pH change leading to subsequent cytotoxicity, they were pre-incubated in growth medium for 2 days. There were no significant increases in pH induced by the scaffolds stored in growth medium (BG, 1P, 2P, and 3P). The growth medium served as a reference. The pH values of the 1P (6.64 ± 0.13) and 2P (6.60 ± 0.09) scaffolds were significantly lower than that of pure medium (7.10 ± 0.10) after 1st day and the pH values of the 1P (6.82 ± 0.19) and 2P (6.83 ± 0.17) scaffolds were significantly lower than that of pure medium (7.26 ± 0.08) after the 2nd day (Figure 4C).

### 3.4. Polymer Infiltration Strengthens Scaffold Structure

The stiffness was measured up to a maximum force of 5000 mN. The force increases until the individual struts break. From this point on, an abrupt drop in force took place. However, since the remaining scaffold remained intact, the curve rose again afterwards. In the non-leached specimens, none were completely destroyed. It could be shown that PLGA infiltration did not lead to any changes in the fracture behavior (linear elastic behavior). However, differences could only be detected after leaching. The leached BG scaffolds showed a nearly horizontal fracture line. This occurred because the glass had a weakened structure due to the dissolution of many ions and the stresses between the gel layer and the non-leached glass. Complete failure happened in a range between 1500 and 3500 mN. At the end of the measurement, only small fragments (crumbs) were present in all three leached BG samples. The fluctuations in the curves can be explained by the fractures of the bars and the subsequent pressure build-up. The leached 3P scaffolds did not appear to have been destroyed at all. The kinks in the curves were due to slight flattening of the top surface. The PLGA supported the scaffold by its elastic properties. No significant differences were observed between the leached 1P, 2P and 3P scaffolds. However, no PLGA-infiltrated scaffold could be destroyed with the maximum possible test load of 5000 mN. Therefore, the stiffness beyond the maximum test force could not be represented (Appendix A).

### 3.5. No Cytotoxic Effects by Polymer Infiltration

The cytotoxicity assay was performed to evaluate if the ion release negatively influenced the cell viability. In accordance with International Standards DIN EN ISO 10993-5 2009-10 norm not only murine L929 fibroblasts, but also the primary pACs and hMSCs (Figure 5) and K4IM (Appendix A) were used since they were included in the following experiments. The normalized cell viability (negative control, which represents the fresh growth medium was set 100%) of all cell types (L929, pACs, hMSCs and K4IM) tested with four different scaffolds variants (BG, 1P, 2P, 3P) was over 95%. All tested substrates (growth medium after 48 h scaffold incubation) were highly significantly more cytocompatible compared to the positive control (DMSO, under 50%), which was highly cytotoxic. Therefore, none of the four scaffolds variants was cytotoxic and all could be used for further cell colonization experiments (Figure 5). 

### 3.6. High Viability over 35 Days Cultivation Period

The viability assay was performed to check the cell viability (pACs, hMSCs) after 1–5 weeks of cultivation on the scaffolds. Identification of the viable cells was possible due to colorless FDA, which is transported through an intact cell membrane and could be hydrolyzed by esterases to fluorescein distributed inside the living cells. PI is a red fluorescence marker, intercalating into the DNA when the nuclear membranes become permeable and shows a nuclear distribution. PI labeled the dead cells. After cell adherence to the scaffold struts, most of the cells were viable during the cultivation time of 35 days (Figure 6). In particular, the colonization of BG and 1P scaffolds with pACs showed a high amount of viable cells on the struts and around the pores. In contrast to that, the amount of viable cells on the 2P and 3P scaffolds was lower and the cells were widely dispersed after 7 days of cultivation as well as after 35 days. Still, after 7 days of cultivation the BG, 1P, 2P and 3P scaffolds were nearly fully colonized by the hMSCs. The struts were covered by densely packed hMSCs up to 35 days. The hMSCs showed a spindle shaped and elongated cell body so that the pores could be covered. The cell shape could be assessed more clearly at a higher magnification (Appendix A). The calculation of the pACs’ viability showed significant differences between the colonized BG scaffolds in comparison to the colonized 2P scaffolds after 7 days. Although there were no further significant differences after 21 days and 35 days, the same trend could be seen as after 7 days (Figure 6I). Interestingly, there was a reduction in viability per hMSC-colonized scaffolds after 7 days, also with a significant difference between BG and 3P scaffolds. After 21 and 35 days, the viability was the highest in the 3P-colonized scaffolds in comparison to the other scaffolds. The viability significantly increased in the 3P scaffolds from 7 to 35 days (Figure 6J). 

The calculated colonized surface of pACs growing on the scaffolds was highest after 21 days in the BG scaffold (2.2 mm^2^). At every time point of analysis, pAC colonized 2P and 3P scaffolds with a lower colonized surface (difference not significant) when compared to their growth on BG and 1P scaffolds (Figure 6K). Interestingly, the highest hMSC-colonized surface was reached after 7 and 21 days on the 2P scaffold (5.6 mm^2^). The colonized surface of the 2P scaffolds was significantly higher than that of the colonized BG and 1P scaffolds after 7 days but also significantly higher than the 2P scaffolds after 35 days. There was also a reduction in colonized 2P scaffold surface from 21 days to 35 days. The same trend could also be seen in the colonized surface of 3P scaffolds, while the colonized surface of BG and 1P scaffolds was constant over the whole colonization time (Figure 6L). The cell seeding efficiency was in all scaffold variants colonized with pACs or hMSCs higher than 70%. No significant differences could be calculated, but in the pACs-colonized scaffolds a slight decrease over time could be noticed, especially in scaffolds with a higher polymer amount (Figure 6M). More hMSCs could attach to the scaffolds in comparison to the pACs. No significant differences could be calculated because the cell seeding efficiency lied over 95% in all scaffold variants, not only after the first 24 h but also after three days of cultivation (Figure 6N). 

### 3.7. Scanning Electron Microscopy Analysis

The cell attachment of pACs and hMSCs on the different scaffolds was visualized with the SEM analysis. After 7 and 35 days, the struts and pore walls of the BG and 1P scaffold were almost completely colonized with pACs. The cells were flattened, spread around the whole struts and covered the pores (Figure 7(A1–B2)). The highly dispersed pACs in the 2P and 3P scaffolds exhibited a round cell shape not only after 7 but also after 35 days (Figure 7 (C1–D2)). The hMSCs showed a long and spindle shaped cell body with a high number of filopodia in all four scaffold types, especially after 7 days. In the 1P scaffold, a synthesis of fibrous ECM could be also seen that covered not only the pores but also the PLGA (Figure 7(F1)). The PLGA could be distinguished from the glass. With increasing PLGA amount, the cells were only able to adhere to the PLGA.

### 3.8. Sulfated Glycosaminoglycan per DNA Contents and Metabolic Activity Depend on Cell Type and Time Point of Culturing

The GAG per DNA amount was determined after 1, 3, 7, 21 and 35 days. The GAG per DNA amount was slightly increasing over time in pAC-colonized scaffolds. At 21 and the 35 days, high standard deviations could be recognized. After 21 and 35 days the polymer-infiltrated scaffolds had a higher GAG per DNA amount than the pAC-colonized scaffolds (Figure 8A). The GAG per DNA amount was also continuously increasing in the hMSC-colonized BG scaffolds, while in the 1P scaffolds the highest GAG per DNA amount could be already recognized after 1 day and was then slightly decreasing. In the 2P and 3P hMSC-colonized scaffolds more variations were seen (Figure 8B). The metabolic activity was evaluated after 7 and 35 days and was related to the uncolonized scaffolds. The pAC-colonized scaffolds showed no significant differences after 7 days. In all four scaffold types only a slight trend of reduced metabolic activity could be seen (Figure 8C). The metabolic activity of hMSC-colonized scaffolds was highest on the BG scaffolds after 7 days (74%). A not significant reduction in metabolic activity could be determined not only on the BG scaffolds but also on the 1P and 3P scaffolds after 35 days. A slight increase in the metabolic activity could be calculated in the 2P scaffold after 35 days compared to 7 days (Figure 8D).

### 3.9. Cartilage Related Protein Expression

Cartilage-specific ECM proteins, such as collagen type II, cartilage proteoglycans (PG) and the chondrogenic transcription factor SOX9 were analyzed by immunocytochemical staining (Figure 9). The pACs and hMSCs were cultivated on the different scaffold variants for 21 days before being immunolabeled for collagen type II (green), PG (red) and the transcription factor SOX9 (green). The ECM proteins were present around, between and within the cells. The protein expression of collagen type II, PG and SOX9 tended to be reduced with a higher amount of PLGA in pAC-colonized scaffold variants (Figure 9I). The collagen type II immunoreactive area per cell of pAC-colonized BG scaffolds (0.0037 ± 0.0027%) was significantly larger in comparison to the pAC-colonized 3P scaffolds (0.00023 ± 0.00094%). The relative expressions of PG and SOX9 were generally higher in BG scaffolds than in PLGA-infiltrated BG scaffolds colonized with pACs (not significant). 

In general, the protein expression per cell of collagen type II and SOX9 was higher in hMSC-colonized scaffolds than in pAC-colonized scaffolds. The hMSC-colonized 2P scaffolds tended to express more proteins in comparison to the other colonized scaffolds variants. No significant differences could be evaluated between the different scaffold variants colonized with hMSCs (Figure 9J). 

### 3.10. Non-Induced Human Mesenchymal Stem Cells Express Cartilage-Specific Genes

Several cartilage-associated genes were analyzed by real-time PCR analysis to estimate the effect of BG and BG with single infiltration with PLGA on pACs and hMSCs after 7 and 21 days of dynamic cultivation and compared to the native porcine cartilage (marked as red line). Gene expression of collagen types II, IX and XI as the most prominent cartilage-specific collagen types was upregulated in the pACs cultures after 21 days in comparison to the 7 day cultivation time point for both scaffold variants (Figure 10A–C). After 21 days the expression was also higher in comparison to the native porcine cartilage. In the hMSCs culture, the expression of collagen type II (except for the BG scaffolds 21 days) (not significantly), IX (highly significantly) and collagen type XI (significantly) was lower in comparison to the native porcine cartilage, but a time dependent increase in the expression could be seen in the colonized BG scaffolds (Figure 10A–C). The transcription factors SOX9 and FOXO1 were not only expressed by the pACs but also by the hMSCs on the BG and 1P scaffolds. In general, the expression was lower (difference not significant, except for 21 days old pure BG scaffolds) in comparison to the native porcine cartilage and pACs showed a higher expression than hMSCs (Figure 10D,E). After 21 days of cultivation time, the expression of aggrecan was upregulated in comparison to the 7 days, not only in the pACs but also in the hMSC culture. After 21 days the expression of aggrecan in the pAC culture was higher (not significant) than in the native porcine cartilage (Figure 10F). Relative gene expression of COMP was significantly lower in both pACs and hMSCs compared with native porcine cartilage, but there was an increasing trend in COMP expression in BG and 1P scaffolds in both cultures over time (Figure 10G). The relative collagen type I gene expression of pAC cultures increased in both scaffold types over time in contrary to the hMSCs, where a decrease over time could be observed with the 1P scaffolds (Figure 10H). The hypertrophic marker collagen type X was significantly lower expressed in the pAC-colonized BG and 1P scaffolds in comparison to the native porcine cartilage. The relative collagen type X expression was slightly reduced in 1P scaffolds seeded with hMSCs (not significant) with increasing time but still higher in comparison to the native porcine cartilage (Figure 10I).

## 4. Discussion

The basic idea behind the invention of bioactive glass was to facilitate bone regeneration [36]. Numerous studies have shown that BG can lead to an increase in the adherence and proliferation of osteoblasts through the hydroxyapatite layer formed [52,53,54], but also that BG stimulated bone formation [55,56,57]. Well-known examples for silicate bioactive glasses are BG1393, BG45S5 and BG60S5 [58]. One shared characteristic of them is the formation of a hydroxyapatite layer by typical conventional BGs [59] which is due to their chemical composition, releasing Ca^2+^ and PO_4_^3−^ ions [59,60]. In some in vitro studies, cartilage regeneration has also been addressed with these glasses [61,62]. However, these BGs were only used as composite material [61,62] and not as “single material” scaffolds like in our study. The fact that BG can be used for cartilage tissue engineering is insufficiently explored so far. 

In a previous study we showed that a highly porous bioactive glass scaffold allows the adhesion, survival and cartilage ECM expression of primary pACs and hMSCs as well as chondrogenesis [49]. Comparing two cell types, this previous study revealed the following results: chondrogenesis of hMSCs was induced by a pure BG scaffold without any chemical chondrogenic supplements, and pACs maintained their chondrocyte-specific phenotype over the whole in vitro culture period of 35 days. Nevertheless, the pure BG scaffold is not sufficiently stable for the mechanical load in the knee joint. An ideal scaffold for cartilage regeneration should resist the mechanical loading during joint movements and provide an appropriate 3D structure for cell survival, ECM formation and nutrient transport [63,64]. For this reason, infiltration with PLGA was performed in the current study to improve the stability of the entire scaffold. A polymer was chosen for BG infiltration, due to its good processability, low reactivity and its biodegradability [65,66]. In addition, previous studies showed cytocompatibility of polymers like PLLA with chondrogenic cells [48,67]. Until now, it was still unclear which amount of PLGA is useful to increase the mechanical stability of the scaffold. As the results showed, the infiltration process with PLGA could be further optimized in future, since the polymer was not homogeneously distributed inside the scaffold because it was a handmade process not automated by devices. However, in this study the basic question of whether an optimal scaffold topology can be achieved has been addressed and, further investigations of BG/PLGA composites will also use established fabrication techniques [48]. The inhomogeneity in PLGA distribution in the BG scaffold did not affect the overall porosity of these composite scaffolds. Further optimization of this technique for more homogeneous infiltration, e.g., under rotatory conditions is a focus of future experiments as well as the opportunity to apply other commonly known techniques such as thermally induced phase separation (TIPS). Indeed, many different protocols and fabrication techniques exist to establish composite scaffolds with synthetic polymers including the TIPS, Fused Deposition Modeling (FDM) and electrospinning (ESP) with cooling system [48,66,68,69], which are costly and time consuming. Therefore, a simple and quick infiltration method was chosen. One important advantage of this method is, that, as supported by the data, the scaffold topology, particularly the porosity, previously proven to be useful for cartilage tissue engineering [49] did not show major changes by polymer integration into the BG scaffold. Porosity and pore interconnectivity are very important not only for the maintenance of the cell phenotype, but also for cell migration into inner parts of the scaffold [67,70]. Despite PLGA infiltration, the total porosity in our scaffolds was still very high (over 76%). As in other studies describing composites of polymers like PLLA, PLGA with BG particles, no significant correlation between polymer/BG amount and porosity was found [71,72,73]. It is unclear whether the inhomogeneous polymer distribution in the scaffold also affects the cells. In particular, the viability and SEM images suggest that increased polymer contents in the composite scaffold may reduce the direct contact of pACs with the BG and thus, adherence may also be impaired. The cell distribution within the pores is not only essential for bone, but also for cartilage regeneration and also the cell morphology provides information about cell-material interactions. The evaluation of cell morphology was performed with SEM, which showed not only the cell adherence at the glass struts as well as at the polymer, but allowed also estimations concerning the cytoskeleton architecture. This is supported by another study that showed the typical elongated cell body with filopodia of adipose tissue-derived mesenchymal stromal cells on BG45S5/PLLA scaffolds [74] or of mouse bone-derived mesenchymal stem cells on BG/PLGA scaffolds [75]. Furthermore, it could be possible that through PLGA infiltration, the pores become too “closed” for chondrocytes and a cell–cell interaction or the supply with suitable nutrients might be reduced. Nevertheless, small cell cluster formation as observed with the pACs on 2P and 3P might not hinder chondrogenesis since mesenchymal cell condensations are known to initiate chondrogenesis during development of cartilage. Even though the distribution of PLGA was not completely uniform, an increase in stability was achieved, primarily before the leaching process. The leaching process applied here is completely different and not comparable with those described in other studies that used a salt leaching method to produce the pores inside biodegradable scaffolds for bone regeneration [71,76,77,78]. Nevertheless, the leaching process of CAR12N performed with HCl is necessary to form a bioactive gel layer around the glass struts and to achieve bioactivity by ion release. During the leaching process in HCl, the 3P scaffolds exhibited a higher pH fluctuation than the pure BG scaffolds, which resulted in a more uniform pH increase over time. Interestingly, the pH of all PLGA-infiltrated scaffolds was always lower than that of BG scaffolds during storage in water (28 days) and was in the range suitable for cell culture. It may suggest that the PLGA prevents an ion release from the BG by covering the gel layer, but there may also be some hydrolytic degradation of the PLGA in acidic lactic acid monomers [66]. In a previous study, it has already been published that PLGA led to a release of acidic degradation products, both in vitro and in vivo [79]. However, the PLGA used in this study has a long degradation profile as reported by the manufacturer (Evonik) of at least 18 months.

Modulation of the slightly alkaline pH associated with the BG by polymers like PLLA presents an advantage of this material combination for applications in the biological system [71]. PLGA/BG45S5 scaffolds compensate the pH of PBS solution [80]. The calculation of the exact amount of ion release was not performed in this study but will be carried out in future. It is important due to the fact that the ion release in a therapeutic dose is responsible for the gel layer formation, the cell adherence on the glass struts and cell proliferation [81,82,83,84,85]. 

The ion release and hence, bioactivity, can be also seen as a therapeutical tool [86,87,88,89] and has possibly stimulatory effects on intracellular calcium spiking, ECM production or cell proliferation, similar to mechanotransduction [90]. Mechanical stimuli, like a dynamic rotation system were used to achieve cell attachment in this study. We could show, in accordance to other studies which had a homogenous cell distribution on the scaffold surface [91], an increase in cell viability [92] and of sGAG production through the continuous nutrition flow in comparison to static cultivation [93]. The mechanical stimuli will be recognized by the primary cilia of chondrocytes, which are non-motile sensory organelles with a few micrometer length [94]. Integrins, especially α2, α3 and β1, accomplish chondrocyte adherence to the ECM and are expressed on the primary cilium [95]. It has been shown that β1-integrins are responsible for adherence to biodegradable polymers, such as PGA, PLA, and PCL, in both, chondrocytes and MSCs [96].

Our current study confirmes a high cyto- and biocompatibility of PLGA functionalized BG scaffolds (1P, 2P, 3P) with both cell types by the cytotoxicity and viability assay not only after 7 and 21 days but also after 35 days of cultivation. Porcine articular chondrocytes were selected due to the fact that they can be harvested from healthy knee joints, their easy and abundant availability, superior proliferation and ECM synthesis capacities. The use of bone marrow-derived hMSCs was chosen because they are easily accessible, have the ability of chondrogenic differentiation and, the problem of immune rejection after implantation, due to low immunogenicity, could be circumvented [97,98]. 

In contrast to pACs, the hMSCs showed not only better adherence to the scaffolds infiltrated multiple times with PLGA, but also maintenance of DNA content and protein expression over the cultivation period. It should be emphasized that the chondrocyte-colonized area of 1P scaffolds was comparable to the colonized area of BG scaffolds, whereas the adherence of pACs on scaffolds decreased with higher PLGA content. This fact led us to the decision to perform further PCR analyses with the BG and 1P scaffold variants. There were discrepancies between colonized area and measured DNA contents probably due to the flattened cell shape of hMSCs. The results show that PLGA does not affect chondrogenesis in pACs. Cell adherence is a key criterion to be fulfilled for a cartilage scaffold and is also related to cell cycle progression and hence, DNA synthesis [99]. It is unclear how many cells adhere inside the scaffold. Unfortunately, conventional cutting techniques undertaken to show the cell distribution and ECM deposition inside the scaffolds failed due to the high glass content and hence, the brittleness of the scaffolds. Since hyaline cartilage consists of about 60% water and 40% ECM, one of the main goals for cartilage tissue engineering is to maintain and increase ECM synthesis [100]. Even though the metabolic activity of hMSCs in multiple times PLGA-infiltrated scaffolds decreased during the cultivation period, significant increases in sGAG synthesis but also in protein expression of collagen type II, cartilage related proteoglycans and the chondrogenic transcription factor SOX9 could be detected. The transcription factor SOX9 is expressed and activated during mesenchymal condensation and in native cartilage tissue. Furthermore, SOX9 is an activator of, for example, collagen type II, IX, aggrecan and matrilin 3 and therefore plays, an important role during early chondrogenesis in 3D cultures [101,102,103,104]. Since there were no major differences in protein expression between the multiple and single-infiltrated BG scaffolds except for a significant decrease in collagen type II synthesis by pACs in 3P scaffolds, gene expression analysis was restricted to BG and 1P. We were able to confirm this relationship between collagen type II and SOX9, although the relative gene expression of SOX9 was not significantly increased over cultivation time. An increase in the relative gene expression of collagen type II and aggrecan was detected in the pACs but also in the hMSC-colonized BG and 1P scaffolds. Another early transcription factor which is very important within the embryonal condensation stage during chondrogenesis, is FOXO1 [105]. The relative expression of FOXO1 per se in undifferentiated hMSCs in the culture indicated that CAR12N probably initiated the chondrogenesis. Some authors found that FOXO1 also contributes to the maintenance of cartilage homeostasis in adult chondrocytes [106,107]. In addition, future investigation will be performed with respect to dedifferentiation and hypertrophic differentiation of chondrocytes using immunocytochemical staining for collagen type I and the hypertrophy markers collagen type X, and matrix metalloproteinase (MMP)13 as well as osteogenic runt-related transcription factor 2 (RUNX2) [108,109] or alkaline phosphatase.

To allow longterm culture we decided to use supplements such as 10% FCS and 5% PL for pAC and hMSC scaffold cultures as well as added amphotericin to inhibit fungal contamination. These additives could impair chondrogenesis or facilitate unwanted hypertrophic differentiation, but still allowed cartilage-specific marker expression in our study suggesting that the BG had sufficient chondro-inductive effects. Due to the fact that the relative gene expression of collagen types I and X tended to be downregulated in longer cultivation time, we hypothesized a beginning of chondrogenesis. Moreover, the upregulation of the relative gene expression of minor collagens like type IX and XI not only in the pACs but especially in the hMSCs indicated that the formation of a stable collagen network and thus the organization of a complex cartilage ECM had begun with a long-term cultivation [110]. Another ECM component, which is responsible for compressive resilience of articular cartilage during joint loading, is the proteoglycan aggrecan [111]. The upregulation of aggrecan and COMP in both, pAC and hMSC cultures and the fact that other scientists have also demonstrated the regulation by mechanical stimuli of these ECM components confirmed our hypothesis regarding the importance of a dynamic cultivation method [112,113,114,115]. It seemed that not only the dynamic cultivation strategy and the porosity of the scaffolds but also the pH has influence on the cross-linking of the collagen chains. This effect of H+ ions on cross-linking has been reported by another group, which showed the influence of BG45S5 on the bone regeneration [116]. 

Based on these data, our hypothesis for further experiments is that these infiltrated PLGA BG scaffolds are might be well suitable for a co-culture model to regenerate cartilage defects. In the bottom region, with a higher amount of PLGA, hMSCs were colonized, while chondrocytes will have direct contact to the pure BG struts in the upper phase of the scaffolds. In addition, co-culture was also advocated to prevent dedifferentiation and instead, promote cartilage-associated ECM formation [117]. 

A degradation of the colonized scaffolds with and without PLGA was not observed during the 35 days of culturing, since size and shape of the scaffolds were maintained. A slow degradation profile is wanted for the scaffolds since neocartilage formation by the chondrogenic cells needs also a long time. Nevertheless, the degradation and the exact release of silicon (Si), sodium (Na), phosphate (P) and boron (B) has to be monitored in detail in a long-term experiment not only in vitro but also in vivo. 

## 5. Conclusions

The present results confirmed that these scaffolds allow the adherence, proliferation, sGAG synthesis and cartilage-specific gene and protein expression. Thus, our CAR12N scaffolds are well suited for cartilage tissue engineering. A detailed scaffold characterization in view of PLGA distribution and effects of PLGA integration on pore perimeters, strut diameters and porosity revealed no significant changes in overall topology, which was suitable for pAC and hMSC adherence. While PLGA infiltration had no major effect on pH changes after leaching, it is of utmost importance for mechanical loading and hMSC-colonization. PLGA-infiltrated scaffolds had a higher stiffness in comparison to the pure BG, which are often regarded as too brittle for future clinical application. Nevertheless, the biomechanical data are preliminary and a detailed comparison of the scaffold with PLGA and native cartilage has to be undertaken. This study demonstrated that chondrogenesis, particularly of hMSCs, and the maintenance of the chondrogenic phenotype of pACs occurred in single PLGA-infiltrated scaffolds without inductive growth factor supplementation. Further work should focus on the gene expression of cartilage related matrix proteins like collagen type II, XI, IX and XI, aggrecan and the transcription factors SOX9 and FOXO1 of pACs and hMSCs on colonized CAR12N/PLGA composite scaffolds after mechanical loading in a bioreactor. Testing the biocompatibility and chondrogenesis in vivo with the pure CAR12N and single PLGA/BG scaffolds are one step further towards clinical application for cartilage defect treatments.

Particularly, the combined implantation of single PLGA-infiltrated CAR12N scaffolds with hMSCs isolated from an iliac crest biopsy during the same surgery into chondral defects could provide a reasonable approach as well as to place a cell-free scaffolds on a defect providing access to subchondral bone marrow.

## 6. Patents

The BG scaffolds (CAR12N) were produced by the patented glass composition (DE 10 2018 114 946 B3, 2019). The scaffold systems were prepared according to our patent DE 10 2019 124 879 A1.

## Figures and Tables

**Figure 1 cells-11-01577-f001:**
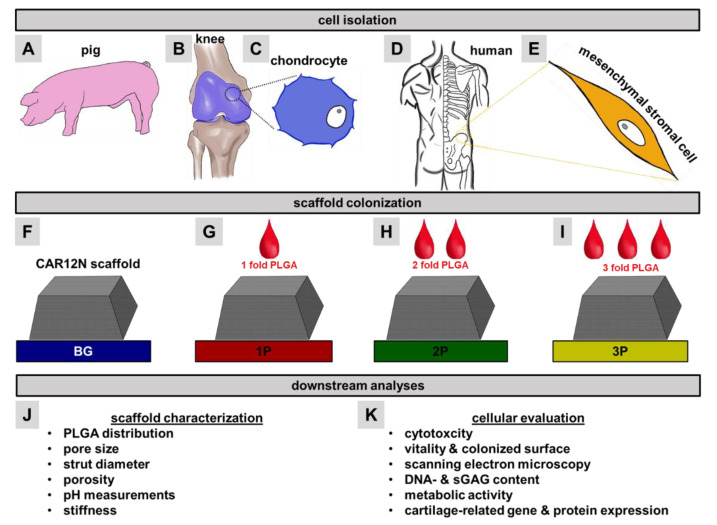
Graphical abstract. Knee joints from the pig (**A**,**B**) were explanted in order to isolate native chondrocytes (**C**). From the human iliac crest (**D**) mesenchymal stem cells (**E**) were obtained. Primary cells were subsequently colonized on pure bioactive glass scaffolds (BG, **F**), which were one-time (1P, **G**), twice (2P, **H**), or three times (3P, **I**) infiltrated with Poly(D-L-lactide-co-glycolide) (PLGA). Prior to colonization extensive analyses were performed to characterize the material properties (**J**). After up to 35 days in a dynamic culture, the cellular changes (cytotoxicity, viability, colonized surface, scanning electron microscopy, DNA and sulfated glycosaminoglycan (sGAG) contents, metabolic activity, immunocytochemical staining and real time polymerase chain reactions (PCR)) were examined in more detail (**K**).

**Figure 2 cells-11-01577-f002:**
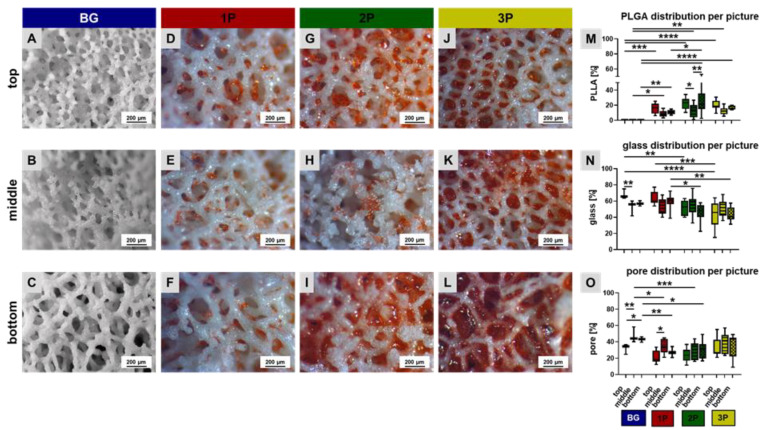
Detailed view of the four scaffold variants. The scaffolds (BG = pure bioactive glass scaffold (**A**–**C**) in blue (**M**–**O**), 1P = single (**D**–**F**) in red (**M**–**O**), 2P = twofold (**G**–**I**) in green (**M**–**O**) and 3P = threefold PLGA infiltrations (**J**–**L**) in yellow (**M**–**O**)), were vertically cut and light microscopic images were taken from the top view (**A**,**D**,**G**,**J**), from the middle (**B**,**E**,**H**,**K**) and the bottom (**C**,**F**,**I**,**L**) view. The percentage distribution of PLGA (**M**), of glass (**N**) and pores (**O**) per picture was calculated. *n* = 4. Ordinary two-way ANOVA with multiple comparison, *p* values: * *p* < 0.05, (**) for *p* < 0.01; (***) for *p* < 0.001 and with (****) for *p* < 0.0001. Scale bar: 200 µm, PLGA was red dyed with the dye Sudan IV red.

**Figure 3 cells-11-01577-f003:**
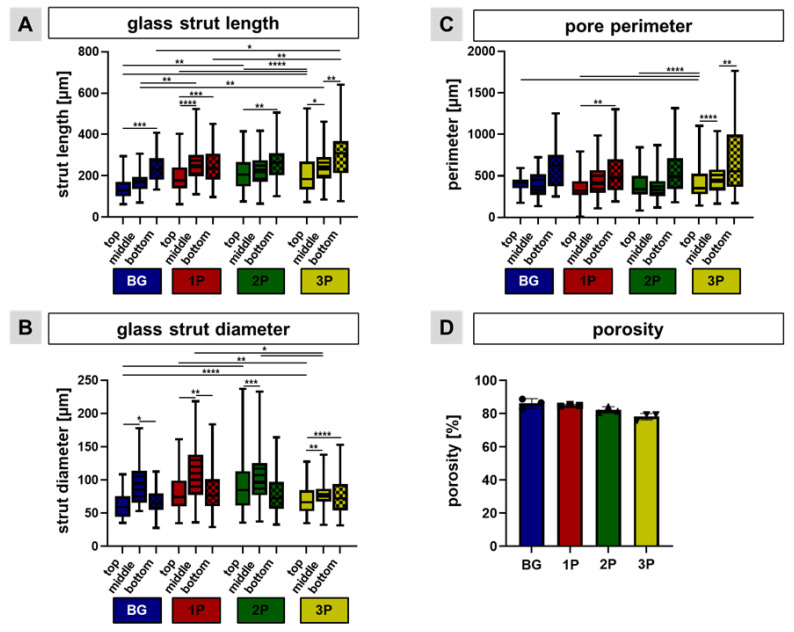
Calculation of scaffold parameters. The glass strut length (**A**), the glass strut diameter (**B**) and the pore perimeter (**C**) were measured in the top, middle and bottom regions of the BG (blue), the 1P (red), the 2P (green) and the 3P (yellow) scaffolds. 1P = single, 2P = twofold, 3P = threefold PLGA infiltrations. Four technical and three biological replicates were included and significant differences were calculated after ordinary two-way ANOVA with multiple comparison (*) for *p* < 0.05; (**) for *p* < 0.01; (***) for *p* < 0.001 and with (****) for *p* < 0.0001. The porosity (**D**) was measured with three different scaffolds from each scaffold variant (BG, 1P, 2P, 3P).

**Figure 4 cells-11-01577-f004:**
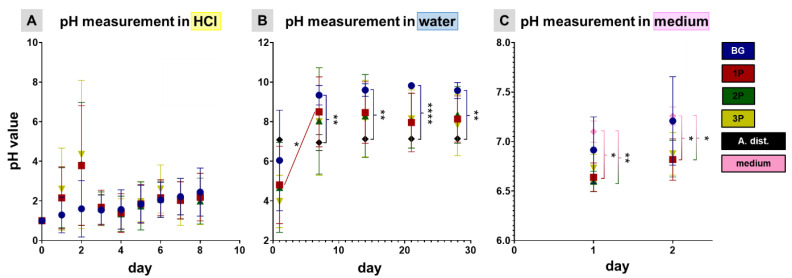
pH measurements in hydrochloric acid (HCl, **A**) in water (**B**) and in cell type specific culture medium (**C**). A total of 100 mg of pure bioactive glass scaffold (BG, blue) of single PLGA(1P, red), twofold (2P, green) and threefold PLGA infiltrations (3P, yellow) scaffolds was stored in 1 mL liquid (HCl, Aqua distilled (A. dist.), 10% FCS containing chondrocyte growth medium). The measurement in growth medium was undertaken only for 2 days, since it is normally changed every 2 days. All measurements were performed at RT (22.02 ± 0.11 °C). Significant differences (from four different scaffold batches) were calculated after two-way ANOVA with multiple comparison (*) for *p* < 0.05; (**) for *p* < 0.01 and (****) for *p* < 0.0001.

**Figure 5 cells-11-01577-f005:**
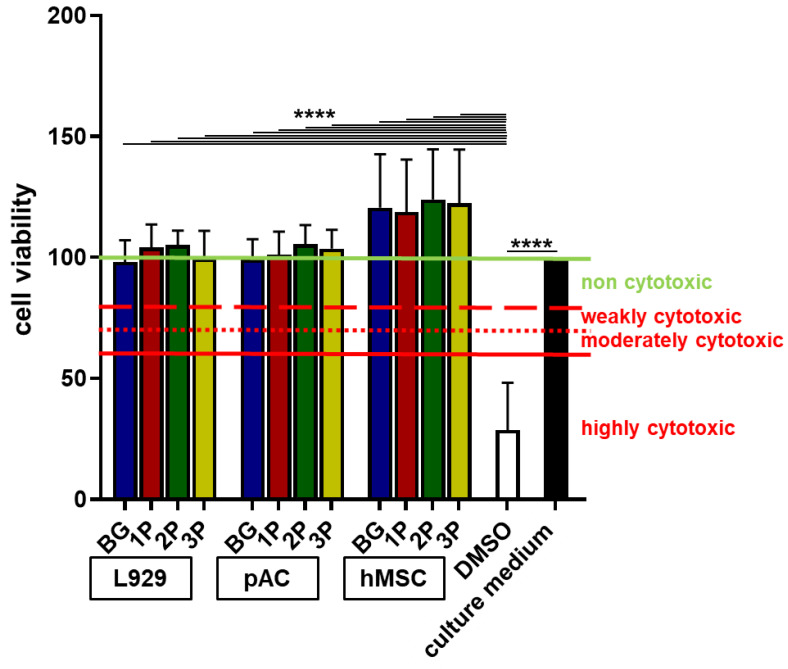
Cytotoxicity Assay using BG (blue bars), 1P (red bars), 2P (green bars) and 3P (yellow bars) scaffold extracts. Viability of L929, porcine articular chondrocytes (pACs) and human mesenchymal stem cells (hMSCs) treated for 24 h with scaffold extracts (48 h extraction) from three different scaffold batches. Cell viability was above 90% in cells treated with all scaffold extracts. Cell viability of the positive control with 10% dimethyl sulfoxide (DMSO, white bar) was below 30% and that of the negative control with culture medium (black) was 100%. The CellTiter 96^®^Aqueous One Solution Cell Proliferation Assay was used to assess cytotoxicity. The pure bioactive glass (BG, blue), single (1P, red), twofold (2P, green) and threefold PLGA infiltrations (3P, yellow). *n* = 3. Ordinary one-way ANOVA with multiple comparison, *p* values: **** *p* < 0.0001.

**Figure 6 cells-11-01577-f006:**
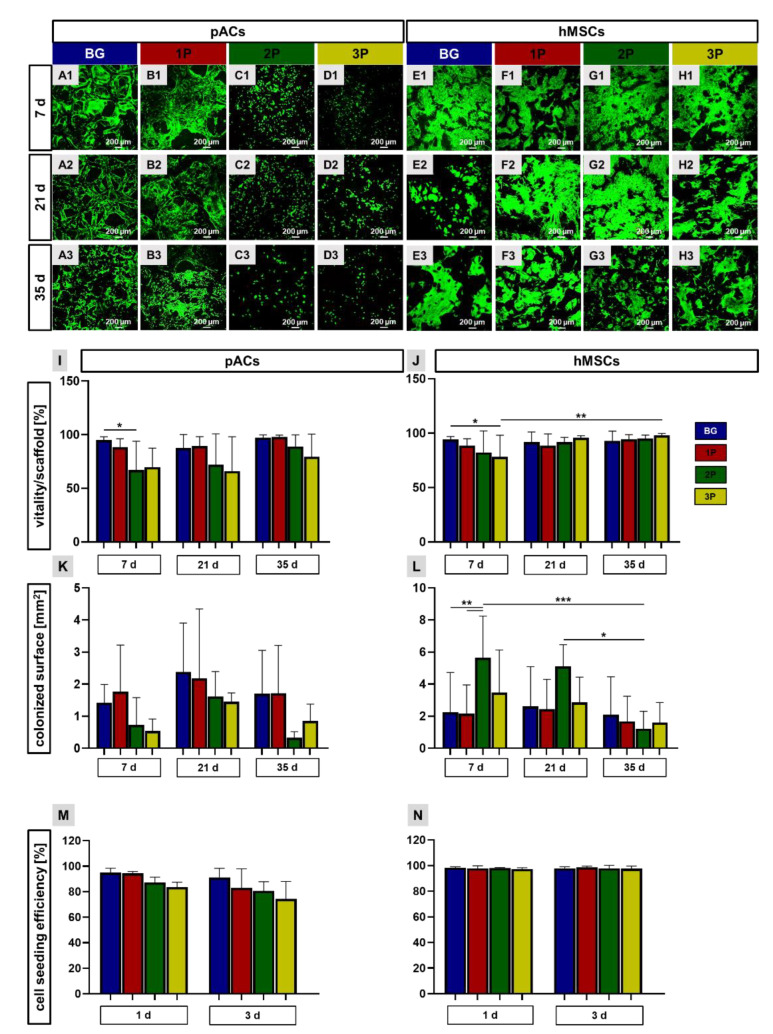
Viability assay of colonized scaffolds, calculation of the cell viability per scaffold surface, the colonized scaffold surface and the cell seeding efficiency. Representative pictures of three independent experiments show vital (green) and dead (red) porcine articular chondrocytes (pACs) (**A1**–**D3**) and human mesenchymal stem cells (hMSCs) (**E1**–**H3**) over a cultivation time of 7 to 35 days (d). pACs (**I**) and hMSCs (**J**) viability per scaffold surface was calculated after 7, 21 and 35 d. Results of the pACs (**K**) and hMSCs (**L**) colonized surface over the cultivation time. The cell seeding efficiency (%) was calculated for pACs (**M**) and hMSCs (**N**) after 1 and 3 d, with no significant differences. The pure bioactive glass (BG, blue), single (1P, red), twofold (2P, green) and threefold PLGA infiltrations (3P, yellow). Three independent experiments (*n* = 3) with cells from three different donors were performed. Two-way ANOVA combined with the post hoc Tukey´s multiple comparison test between the groups: *p* values: * *p* < 0.05, ** *p* < 0.01 and *** *p* < 0.005. Scale bar: 200 µm.

**Figure 7 cells-11-01577-f007:**
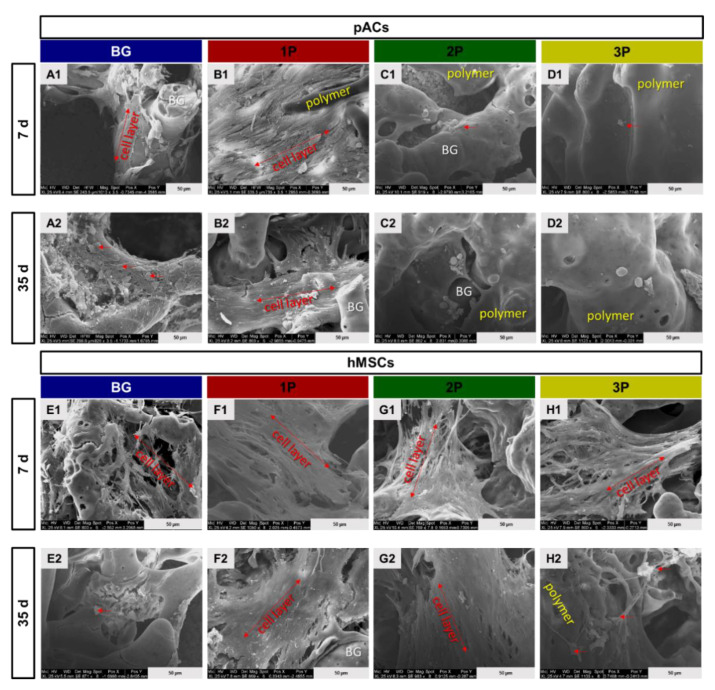
Scanning electron microscopy (SEM) of colonized scaffolds. Representative images of the morphology not only of the pACs (**A1**–**D2**) on the scaffolds after 7 days (d) (**A1**–**D1**) and after 35 d (**A2**–**D2**) but also of hMSCs (**E1**–**H1**) on the scaffolds after 7 (**E1**–**H1**) and 35 d (**E2**–**H2**) in dynamic culture. The pure bioactive glass (BG, blue), single (1P, red), twofold (2P, green) and threefold PLGA infiltrations (3P, yellow). Scale bar: 50 µm. pACs = porcine articular chondrocytes, hMSCs = human mesenchymal stem cells, BG = bioactive glass. Single cells are marked with red arrow and cell layers are marked by double headed arrows.

**Figure 8 cells-11-01577-f008:**
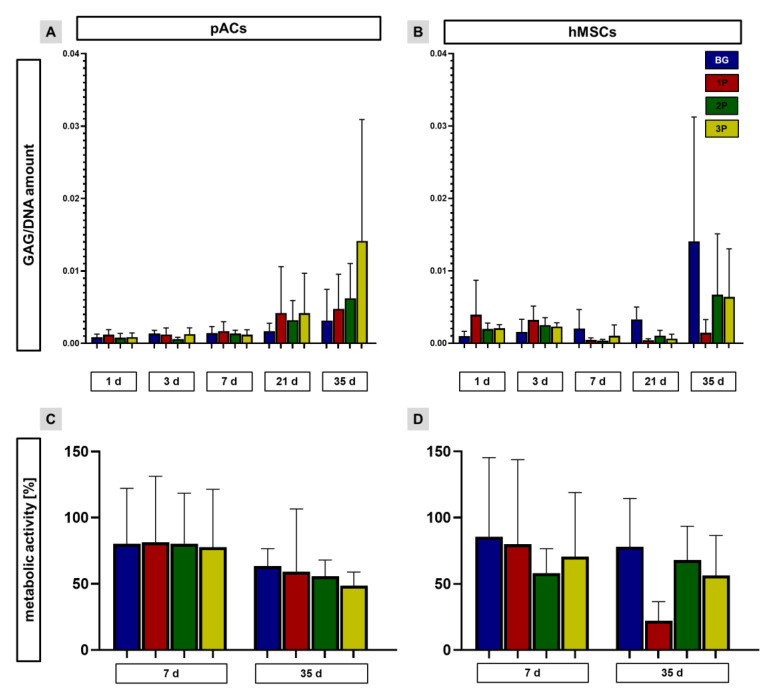
Calculation of the sulphated glycosaminoglycan (sGAG) per DNA amount per scaffolds after 1, 3, 7, 21 and 35 days and the metabolic activity of the cells on the scaffolds after 7 days and 35 days. GAG per DNA amounts of pAC- (**A**) and hMSC-(**B**) colonized scaffolds were calculated after 1, 3, 7, 21 and 35 days (d). The metabolic activity of pAC- (**C**) and hMSC- (**D**) colonized scaffolds was measured after 7 and 35 days and was compared to the color change in the assay caused by uncolonized scaffolds. The pure bioactive glass (BG, blue), single (1P, red), twofold (2P, green) and threefold PLGA infiltrations (3P, yellow). Three independent (*n* = 3) experiments with cells from three different donors were performed. Two-way ANOVA (post hoc Tukey Test) for comparison between the groups. No significances pACs = porcine articular chondrocytes and hMSCs = human mesenchymal stem cells.

**Figure 9 cells-11-01577-f009:**
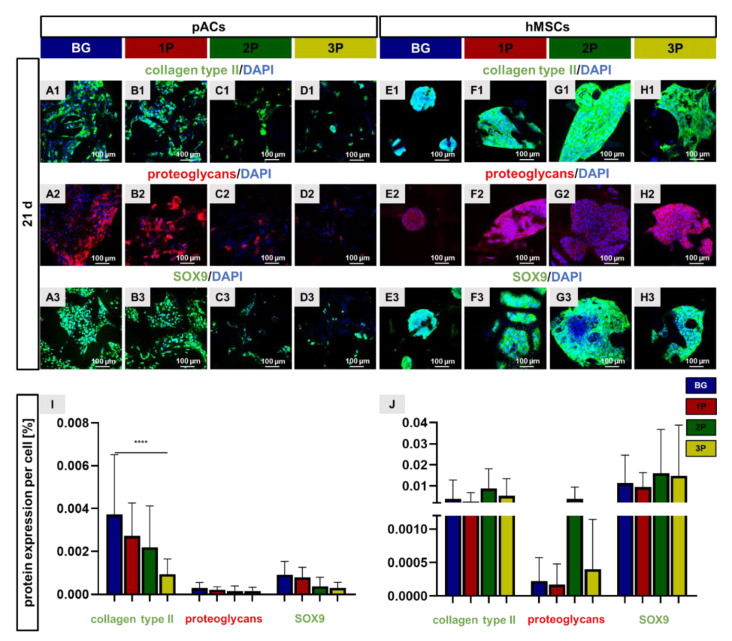
Cartilage-specific protein expression after 21 days of scaffold colonization. Representative immunocytochemical staining of collagen type II (green) and cell nuclei (blue) of pACs (**A1**–**D1**) and hMSCs (**E1**–**H1**), of cartilage proteoglycans (red) and cell nuclei (blue) of pACs (**A2**–**D2**) and hMSCs (**E2**–**H2**) and the transcription factor SOX9 (green) and the cell nuclei (blue) of pACs (**A3**–**D3**) and hMSCs (**E3**–**H3**). The calculation of the immunoreactivity (collagen type II, proteoglycans and SOX9) per cell was performed for pACs (**I**) and hMSCs (**J**). The pure bioactive glass (BG, blue), single (1P, red), twofold (2P, green) and threefold PLGA infiltrations (3P, yellow). Scale bar: 100 µm. *n* = 3. One-way ANOVA (post hoc Tukey Test) for comparison between the groups. *p* values: **** *p* < 0.0001. pACs = porcine articular chondrocytes and hMSCs = human mesenchymal stem cells.

**Figure 10 cells-11-01577-f010:**
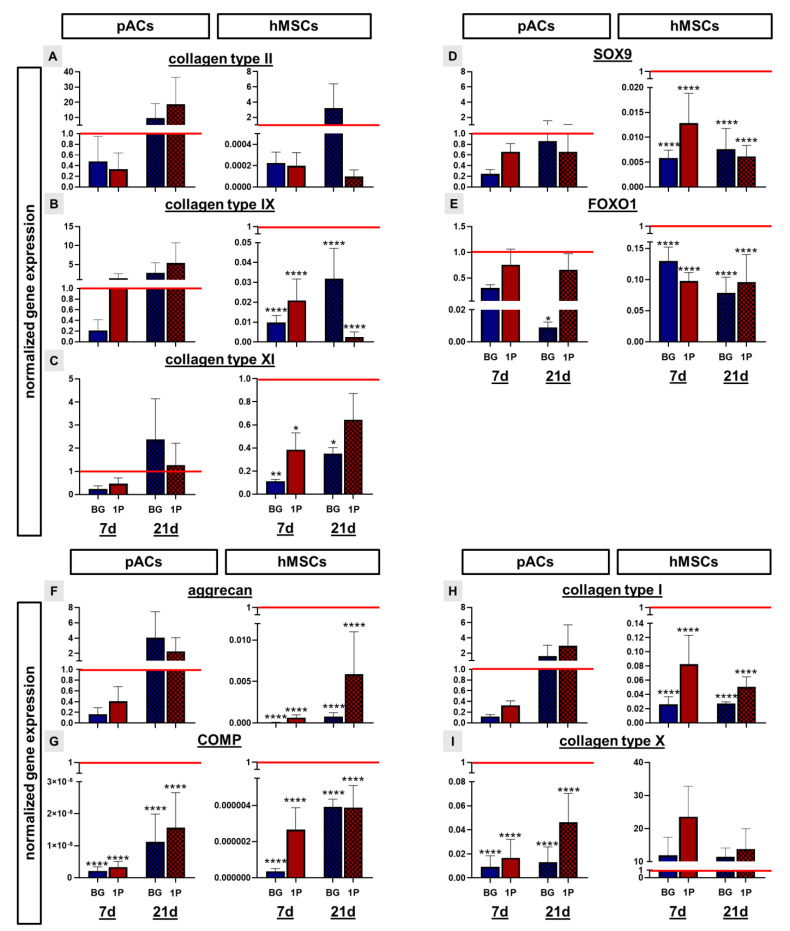
Normalized gene expression of cartilage extracellular matrix components and the transcription factors SOX9 and FOXO1 after 7 (full bars) and 21 (checkered bars) days (d) in the BG scaffolds (blue) and the PLGA-infiltrated BG scaffolds (red). Relative gene expression of collagen types II (**A**), IX (**B**), XI (**C**), SOX9 (**D**), FOXO1 (**E**), aggrecan (**F**), COMP (**G**), collagen types I (**H**) and X (**I**) were related to the native porcine cartilage (red line = 1). The pure bioactive glass (BG, blue) and single (1P, red) PLGA infiltration. Three independent experiments with cells of three different donors are summarized and show mean with standard deviation. Significant differences were calculated after one-way ANOVA (post hoc Tukey Test): *p* values * *p* < 0.05, ** *p* < 0.01, **** *p* < 0.0001. Gene expression was normalized to the reference gene β-actin (ACTB). pACs: porcine articular chondrocytes, hMSCs: human mesenchymal stem cells.

**Table 1 cells-11-01577-t001:** chemical composition of BG.

Chemical	Patented Mass [%]
Silicon dioxide (SiO_2_)	62.7
Sodium oxide (Na_2_O)	25.6
Diphospho pentoxide (P_2_O_5_)	7.0
Boron oxide (B_2_O_3_)	2.0
Potassium oxide (K_2_O)	1.0
Potassium nitrate (KNO_3_)	0.5
Sodium sulfate (Na_2_SO_4_)	0.4
Sodium chloride (NaCl)	0.3
Sodium fluoride (NaF)	0.2
Titanium oxide (TiO_2_)	0.12
Yttria stable Zirconia (ZrO_2_ + Y_2_O_3_)	0.1
Zinc oxide (ZnO)	0.06
Copper oxide (CuO)	0.02
total	100.00

**Table 2 cells-11-01577-t002:** Antibodies used in this study.

Target	Primary Antibody	Dilution	Secondary Antibody	Dilution
type II collagen	Rabbit anti human, Acris Laboratories, Hiddenhausen, Germany	1:50	donkey-anti-rabbit, Alexa Fluor 488, Invitrogen	1:200
cartilage proteoglycans	Mouse anti human, Chemicon International, CA, USA	1:70	Donkey-anti-mouse; Cy3, Invitrogen	1:200
SOX9	Rabbit anti human, Merck, Darmstadt, Germany	1:100	donkey-anti-rabbit, Alexa Fluor 488, Invitrogen	1:200

**Table 3 cells-11-01577-t003:** Detailed information of primers used in the present study.

Gene Symbol	Species	Gene Name	Efficacy	Amplicon Length (Base Pairs)	Assay ID ^#^
ACAN	*Homo sapiens*	Aggrecan	1.95	93	Hs00202971_m1
ACAN	*Sus scrofa*	Aggrecan	1.69	60	Ss03374823_m1
ACTB	*Homo sapiens*	β-actin	1.89	171	Hs99999903_m1
ACTB	*Sus scrofa*	β-actin	1.71	77	Ss03376081_u1
Col1A1	*Homo sapiens*	Type I collagen	2.06	66	Hs00164004_m1
Col1A1	*Sus scrofa*	Type I collagen	1.53	74	Ss03373340_m1
Col2A1	*Homo sapiens*	Type II collagen	2.06 (1.9 *)	124	Hs00264051_m1
Col9A3	*Homo sapiens*	Type IX collagen	1.99 ^#^	52	Hs00951243_m1
Col9A3	*Sus scrofa*	Type IX collagen	1.92	70	Ss06885389_m1
Col10A1	*Homo sapiens*	Type X collagen	1.75	76	Hs00166657_m1
Col10A1	*Sus scrofa*	Type X collagen	1.87	85	Ss03391766_m1
Col11A1	*Homo sapiens*	Type XI collagen	1.80 ^#^	98	Hs00266273_m1
Col11A1	*Sus scrofa*	Type XI collagen	1.74	86	SS03373534_m1
COMP	*Homo sapiens*	Cartilage oligomeric matrix proteine	2.21#	101	Hs00164359_m1
COMP	*Sus scrofa*	Cartilage oligomeric matrix proteine	1.76	117	Ss03375728_u1
FOXO1	*Homo sapiens*	Forkhead box O 1	1.84 ^#^	103	Hs 00231106_m1
FOXO1	*Sus scrofa*	Forkhead box O 1	1.56	107	Ss03388140_s1
SOX9	*Homo sapiens*	SOX9	1.92	102	Hs00165814_m1
SOX9	*Sus scrofa*	SOX9	1.57	145	Ss03392406_m1

all primers from Applied Biosystems ^®^ (life technologies TM). * Primer efficacy determined for porcine chondrocytes. ^#^ primer efficacy determined for human chondrocytes.

## Data Availability

Not applicable.

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
