# Peer review of "Biodegradable Poly(D-L-lactide-co-glycolide) (PLGA)-Infiltrated Bioactive Glass (CAR12N) Scaffolds Maintain Mesenchymal Stem Cell Chondrogenesis for Cartilage Tissue Engineering"

_cells, 2022, doi:10.3390/cells11091577_

Round 1

Reviewer 1 Report

Peer-Review Cells – 1639117

The manuscript entitled “Biodegradable Poly(D-L-lactide-co-glycolide) (PLGA) infiltrated bioactive glass (CAR12N) scaffolds enhances mesenchymal stromal cell chondrogenesis for cartilage tissue engineering” by Clemens Leo Gögele et al. constitutes a very interesting and a well-design study on the enhancement of the stability, physical properties and biological performance of bioactive glass (BG) scaffolds. The manuscript is sufficiently organized and fits within the scope of the journal Cells (ISSN 2073-4409), particularly within the Special Issue “Cell Therapies in Orthopaedics”. However, some issues must be addressed before its consideration for publication.

Major issues:

  1. The novelty of the study is the use of a newly patented BG (CAR12N) in a scaffold for cartilage TE or also the method for PLGA addition via solvent phase infiltration? The authors should clarify this.

  1. (Page 6, subsection 2.9): Please clarify: “A live/dead staining solution was mixed with PI” What was the content of the live/dead staining solution? Calcein AM? Ethidium bromide?

  1. The authors should provide information/discuss the effects of the different PLGA infiltration on the whole scaffold biodegradability in vitro.

  1. The authors need to provide a deeper description of the scaffolds mechanical properties (e.g. compressive modulus).

  1. Please clarify the metabolic activity evaluation. If the uncolonized scaffolds were defined as 100% metabolic active, the lower (<100%) values showed in Figure 9E and F mean that the cell-seeded scaffolds are less metabolic active than uncolonized ones (which is strange). Please change both the Materials and Methods and Results section accordingly.

  1. The error bars in graphs from Figure 9 A, B (DNA content) and 9 E, F (Metabolic activity) are considerably high, which makes extrapolation of conclusions difficult. The authors should look for any outliers in the data analysis or add more experiments.

  1. The authors should provide images with higher magnifications correspondent to the cell images from Figures 7 A1-H3 and Figure 10 A1-H3 to allow a better visualization of cell morphology. These can be added as Supplementary material.

  1. Why the RT-qPCR analysis was just performed for the BG and 1P scaffold groups? What was the rationale to select the 1P group in comparison to 2P and 3P? These should be specified in the main manuscript.

  1. In the RT-qPCR analysis, the expression of bone related genes such as RUNX2 and alkaline phosphatase should be done to confirm that the scaffold is not promoting MSC osteogenesis. This is particularly important in this case because the scaffolds are mainly composed by BG.

  1. Do the authors envisage any possible future strategy to achieve a more homogeneous PLGA distribution within the scaffold structure?

Other issues:

  1. (Page 2, line 90): Poly (glycolic acid) abbreviation is normally “PGA”. Please correct.
  2. Please standardize the way of presenting the Figure captions.
  3. (Page 6, line 215): It is better to use “MSC growth medium” than “stem cell growth medium”.
  4. (Page 6, lines 230, 231): The authors need to specify the rationale for scaffold/extraction medium used.
  5. (Page 6, line 236): Please add the word “medium” to obtain “(...) 100 μL extraction medium”.
  6. Throughout the manuscript when referring to cells, it is more appropriate to use the term “viability” than vitality.
  7. (Page 7, subsection 2.11, line 271): Please replace “fixed” by “stained” to obtain: “(...) stained in 1% osmium tetroxide (OsO4)(...)”
  8. (Page 7, subsection 2.11, line 274): Please replace “à 15 min” by “for 15 min each”.
  9. (Page 7, subsection 2.12, line 296): Why the centrifugation specifically at 16483.55 g?
  10. (Page 8, subsection 2.13): Please clarify which cell viability assay was used. CellTiter-Blue (Promega) or Alamar Blue (ThermoFisher Scientific)?
  11. (Page 8, subsection 2.14, line 321): Please change the word order to obtain a more appropriate form: “After the cultivation time (21 days) scaffolds were fixed (...)”
  12. (Page 8, Table 2 caption): Use capital letter in “Antibodies”.
  13. (Pages 9-10, Table 3 caption): Use capital letter in “Detailed”.
  14. (Pages 9-10, Table 3): Why Col2A1 just for Homo sapiens?
  15. (Page 11, subsection 3.1., lines 382-384): Please complete/reprase the sentence “(...) gave an overview...
  16. (Page 13, subsection 3.3., line 451): Please use “pyrogen-free”.
  17. (Page 16, subsection 3.6., line 526): Please replace “HMSCs” by “hMSCs”.
  18. (Page 19, subsection 3.8., line 580): It is more appropriate to replace the word “downregulation” by “decrease”.
  19. (Page 20, subsection 3.9., line 618): Please correct to the plural form: “ The relative expressions of PG and SOX9 were generally (...)”
  20. (Page 23, line 697): The authors mean PLLA or PLGA?
  21. (Page 23, line 703): Please correct the verbal form to: ”(...) has been addressed (...)”
  22. (Page 23, line 746): The authors mean PLLA or PLGA?
  23. (Page 24, lines 756-759): The authors should complete or rephrase the sentence to improve clarity.
  24. (Page 25, line 793): Please add the word “to” in: “We were able to confirm this relationship (...)”

Author Response

Dear Editor,                                                                                       Nuremberg, 6th April 2022

The authors would like to thank the reviewer for carefully reading the manuscript and very valuable comments. We modified the manuscript according to the reviewer suggestions with a list of changes shown below (attached). All changes performed are indicated in red in the revised version of the manuscript. We improved the English now. We hope you will find this manuscript suitable for publication in “Cells”. Please do not hesitate to contact me anytime for questions regarding this manuscript.

Sincerely,

Univ.-Prof. Dr. Gundula Schulze-Tanzil

(corresponding author)

Reviewer 1

The manuscript entitled “Biodegradable Poly(D-L-lactide-co-glycolide) (PLGA) infiltrated bioactive glass (CAR12N) scaffolds enhances mesenchymal stromal cell chondrogenesis for cartilage tissue engineering” by Clemens Leo Gögele et al. constitutes a very interesting and a well-design study on the enhancement of the stability, physical properties and biological performance of bioactive glass (BG) scaffolds. The manuscript is sufficiently organized and fits within the scope of the journal Cells (ISSN 2073-4409), particularly within the Special Issue “Cell Therapies in Orthopaedics”. However, some issues must be addressed before its consideration for publication.

Response (R): We would like to thank the Reviewer for taking the time to go through our manuscript very carefully and in detail.

Major issues:

  1. The novelty of the study is the use of a newly patented BG (CAR12N) in a scaffold for cartilage TE or also the method for PLGA addition via solvent phase infiltration? The authors should clarify this.

R: The fact that it was possible to distribute a polymer within a pure glass scaffold in the described manner is novel. In most publications describing glass/polymer combinations the main component is a polymer and then glass particles were added. In our case, however, we have a highly porous glass scaffold and it was possible to introduce a kind of polymer sponge into it. However, the small amount of polymer enhanced the scaffold stability and had impact on cellular effects.

In lines 95-97 the novelty of our approach is explained now.

  1.  (Page 6, subsection 2.9): Please clarify: “A live/dead staining solution was mixed with PI” What was the content of the live/dead staining solution? Calcein AM? Ethidium bromide?

R: The authors thank the reviewer for this comment and the reviewer is absolutely right, it was missunderstandable. We clarified this and rewrote the sentence: „The live/dead staining solution consisted of 1 µl propidium iodide (it was not ethidium bromide, we use in our lab propidium iodide) and 5 µL FDA.” Lines 264-268.

  1. The authors should provide information/discuss the effects of the different PLGA infiltration on the whole scaffold biodegradability in vitro.

R: The authors would like to thank the reviewer for this important remark, however, at the moment we are not yet able to give any precise information on the biodegradation of the polymer-infiltrated scaffolds. From the manufacturer Evonik we know that at least 18 months stability can be expected for the pure polymer (PLGA). This is added in lines 841-843 now. In lines 838-839 it is discussed that PLGA influences the pH which could also play a role during degradation. Currently, experiments with the ICP method are performed to determine the individual ions that are dissolved out over time but this method is also very time consuming, hence, it is not possible to add these data within the time frame given for this revision. Of course, the biodegradation in vitro and in vivo is part of an ongoing study in order to compare it with conventional bioglasses. First investigations of in vivo degradability are already underway using subcutaneous implantation in the nude mouse model. Nevertheless, so far we can say that there is continuous ion leakage (see below for the pure BG scaffolds). The scaffolds are therefore theoretically stable in vitro for several months and degradation sets in only very slowly. This is wanted, since stable neocartilage formation needs also months. Degradation is discussed in lines 935-938.

The release of three representative ions (si: silicium, P: phosphate and B: bor) was measured after immersing the leached scaffolds (5 d in 0.1 M HCl) in water for 25 days (Bachelor thesis (Engineering) No. 667, Ms. Vera Kerling, Technical University Georg Simon Ohm, Nuremberg, 2019). It showed a nearly linear release profile over time.

  1. The authors need to provide a deeper description of the scaffolds mechanical properties (e.g. compressive modulus).

R: The authors thank the reviewer for the suggestion, however, it must be mentioned here, that this was an initial biomechanical testing series. The pure BG and PLGA infiltrated scaffolds exposed to pressure did not break similarly, probably due to different surface properties. Hence, exact calculations are very difficult and comparison to measurements with other methods might also be limited. Hence, we prefer not to calculate the compressive modulus based on the data acquired with the device used (GT200 particle compression tester).

Please clarify the metabolic activity evaluation. If the uncolonized scaffolds were defined as 100% metabolic active, the lower (<100%) values showed in Figure 9E and F mean that the cell-seeded scaffolds are less metabolic active than uncolonized ones (which is strange). Please change both the Materials and Methods and Results section accordingly.

R: We thank the reviewer, there was a mistake, the metabolic activity of the uncolonized scaffold was set to zero activity (line 348). Hence, the metabolic activity of all colonized scaffolds was of course higher than that of the uncolonized ones.

  1. The error bars in graphs from Figure 9 A, B (DNA content) and 9 E, F (Metabolic activity) are considerably high, which makes extrapolation of conclusions difficult. The authors should look for any outliers in the data analysis or add more experiments.

R: Figure 9 was changed due to the inclusion of additional time points (1 d and 3 d) and the calculation of GAG per DNA content. An outlier test (Rout) was performed but no outliers could be detected.

  1. The authors should provide images with higher magnifications correspondent to the cell images from Figures 7 A1-H3 and Figure 10 A1-H3 to allow a better visualization of cell morphology. These can be added as Supplementary material.

R: The authors showed now Figure 7 in a higher magnification as supplemental Figure 1.

We are very sorry, but Figure 10 is already in a higher magnification shown, we could not perform pictures at 400x magnification with the Laser scanning microscope due to the fact that the scaffolds are too high and not plan enough to get in the right focus level.

  1. Why the RT-qPCR analysis was just performed for the BG and 1P scaffold groups? What was the rationale to select the 1P group in comparison to 2P and 3P? These should be specified in the main manuscript.

R: We chose BG and 1P for the PCR analyses because we saw in the previous experiments, particularly on the protein expression level that there were no glaring differences due to additional polymer infiltration (2P and 3P). The additional work involved in the preparation is out of proportion to the output and therefore, we will continue to work with the 1P variant in the future. Furthermore, the colonization tests with the pACs on 2P and 3P were not satisfactory and it was often impossible to isolate sufficient RNA. This decision is explained in lines 894-898 now.

  1. In the RT-qPCR analysis, the expression of bone related genes such as RUNX2 and alkaline phosphatase should be done to confirm that the scaffold is not promoting MSC osteogenesis. This is particularly important in this case because the scaffolds are mainly composed by BG.

R: We agree with the reviewer's suggestion to further investigate osteogenic markers. The used BG CAR12N does not contain calcium ions and hence, the risk of bone formation is theoretically limited. The gene expression of the hypertrophy marker collagen type X did not increase in cells cultured on CAR12N. Hence, we have selected a whole range of cartilage-specific markers and will focus more on osteogenic markers in our next project, as we currentlyperform a direct comparison of the cell responses between a conventional (BG1393) and our bioglass (CAR12N). We therefore, ask for your indulgence that further PCR analyses cannot be performed at the moment. This outlook is given in lines 908-914.

  1. Do the authors envisage any possible future strategy to achieve a more homogeneous PLGA distribution within the scaffold structure?

R: We will try to optimize the infiltration technique e.g. by performing it under rotatory conditions and to think about testing other commonly known techniques such as thermo-induced phase separation for polymer integration into the BG scaffolds. This topic is addressed in lines 797-800 now. The advantage of the infiltration technique is that it is be inexpensive and quickly feasible.

Other issues:

  1. (Page 2, line 90): Poly (glycolic acid) abbreviation is normally “PGA”. Please correct.

R: We agree with the reviewer but we stick to the specifications of the manufacturer Evonik and would like to keep the desination PLGA.

  1. Please standardize the way of presenting the Figure captions.

R: The usage of capital letters and blanks was adapted as well as the description of 1-3P. Formatting was adapted.

(Page 6, line 215): It is better to use “MSC growth medium” than “stem cell growth medium”.

R: We have changed it (page 6, line 220).

  1. (Page 6, lines 230, 231): The authors need to specify the rationale for scaffold/extraction medium used.

R: The ratio was 100 mg Scaffold weight in 1 mL liquid (in this case it was cell specific culture medium). We added this sentence (lines 241-242). Thank you.

  1. (Page 6, line 236): Please add the word “medium” to obtain “(...) 100 μL extraction medium”.

R: Done. Now in line 245.

  1. Throughout the manuscript when referring to cells, it is more appropriate to use the term “viability” than vitality.

R: We used the term “viability” throughout the entire manuscript now.

  1. (Page 7, subsection 2.11, line 271): Please replace “fixed” by “stained” to obtain: “(...) stained in 1% osmium tetroxide (OsO4)(...)”

R: We would prefer the term “contrasted”. Now in line 296.

  1. (Page 7, subsection 2.11, line 274): Please replace “à 15 min” by “for 15 min each”.

R: We corrected it. Now in line 299.

  1. (Page 7, subsection 2.12, line 296): Why the centrifugation specifically at 16483.55 g?

R: Well, we preferred the documentation in „g“ as a stadardized method. The g value was determined by converting rpm (rounds per minute).

  1. (Page 8, subsection 2.13): Please clarify which cell viability assay was used. CellTiter-Blue (Promega) or Alamar Blue (ThermoFisher Scientific)?

R: We added the company Promega GmbH to define it in detail. Now in line 343.

  1. (Page 8, subsection 2.14, line 321): Please change the word order to obtain a more appropriate form: “After the cultivation time (21 days) scaffolds were fixed (...)”

R: Thank you. We corrected it. Now in line 352.

  1. (Page 8, Table 2 caption): Use capital letter in “Antibodies”.

We corrected it.

  1. (Pages 9-10, Table 3 caption): Use capital letter in “Detailed”.

R: We corrected it.

(Pages 9-10, Table 3): Why Col2A1 just for Homo sapiens?

R: We found out that human collagen type II had a better binding efficacy than the porcine primer purchased, therefore, we decided to use the human one for both, human MSCs and porcine articular chondrocytes.

  1. (Page 11, subsection 3.1., lines 382-384): Please complete/reprase the sentence “(...) gave an overview...

R: We completed the sentence. „The evaluation of the percentual distribution of PLGA, glass and pores per pictures gave an overview of their position within the scaffold and therefore, the scaffold characteristics. (lines 422)

  1. (Page 13, subsection 3.3., line 451): Please use “pyrogen-free”.

R: Yes, we corrected it. Now in line 491.

  1. (Page 16, subsection 3.6., line 526): Please replace “HMSCs” by “hMSCs”.

R: We rewrote it in this manner: „The hMSCs…..“ because it is the start of a sentence that´s by we wrote it in capital letter. Now in line 587.

  1. (Page 19, subsection 3.8., line 580): It is more appropriate to replace the word “downregulation” by “decrease”.

R: Due to the overall changes in this passage “downregulation” was deleted.

  1. (Page 20, subsection 3.9., line 618): Please correct to the plural form: “ The relative expressions of PG and SOX9 were generally (...)”

R: We corrected it. Now in line 701.

  1. (Page 23, line 697): The authors mean PLLA or PLGA?

R: As it is written in the sentence „PLLA“ is correct in the given context.

  1. (Page 23, line 703): Please correct the verbal form to: ”(...) has been addressed (...)”

R: We corrected it. Now in line 794.

  1. (Page 23, line 746): The authors mean PLLA or PLGA?

R: PLLA is the correct written word in the given context.

  1. (Page 24, lines 756-759): The authors should complete or rephrase the sentence to improve clarity.

R: We hope that we could clarify the sentence now.

  1. (Page 25, line 793): Please add the word “to” in: “We were able to confirm this relationship (...)”

R: Thank you, we added it. Now in line 897.

Reviewer 2 Report

In the present manuscript, the authors compared pure bioactive glass (BG) scaffolds and PLGA-infiltrated BG as potential biomaterials in cartilage repair. They demonstrated that the scaffolds are non-cytotoxic and promote chondrogenesis of porcine articular chondrocytes (pAC) or human mesenchymal stem cells (hMSC). Overall, this is a conclusive preliminary study, which provided a general characterization of the scaffolds. In the following, the reviewer would like to ask some questions and make suggestions to further improve the manuscript.

General aspects:

Line 55: Here, the authors refer to a special form of OA, posttraumatic osteoarthritis, this term should be used.

Line 59: Could the authors please specify, whether MACI includes both in vivo and in vitro infiltration of cells? To the reviewer’s knowledge, cells are seeded/ encapsulated in vitro, thus cell-bearing matrices are implanted.

Line 523: “not that high” = lower, reduced?

Line 312: The authors use both names “CellTiter-Blue” and “alamarBlue” assay. It is true that both assays are based on the same principle and component, namely resazurin. However, to the knowledge of the reviewer, these are different brands/ assays. Could the authors please check whether this is correct? Could the authors please explain the difference between CellTiter-Blue (metabolic activity), and the CellTiter96 (cytotoxicity assay; line 510)?

Minor correction:

1.) Culture/ growth medium for chondrocytes and MSC contained FCS and amphotericine. Both have impairing effects on chondrogenesis and proliferation of cells. It is very unlikely that cells undergo chondrogenic differentiation in presence of FCS (even 1% of FCS is inhibitory). The authors need to discuss that. Why is there no suppression? Moreover, the authors describe that BG is usually used in bone regeneration. Usually, MSC undergo chondrogenesis and further differentiate into osteoblast lineage. Why didn’t the MSC undergo osteogenic differentiation in this study? In particular in presence of FCS, it is very unlikely that a non-hypertrophic differentiation can take place.

2.) The authors explain in the introduction the burdens of MACI, which includes senescence and dedifferentiation of primary cells during in vitro expansion. Therefore, the passage number needs to be provided (maybe the reviewer missed this information) and the dynamics of the de- and re-differentiation would increase the significance of the study. So far, the authors compare the cells at day 7, 21 and 35 to native cartilage. But what was the differentiation status at the time of seeding (after in vitro expansion)? One might assume that there was further dedifferentiation during the earlier time points after seeding on the scaffolds (within the first 7 days), indicated by proliferation. However, at the later timepoints cells might undergo chondrogenesis, as demonstrated by COL2, ACAN etc. expression.

3.) 10% DMSO was used as a positive control of cell death. Is this a suitable control? Usually, TNF plus cycloheximide or comparable components are used to study cell death. Why have the authors decided for DMSO?

4.) The main challenge in in vitro colonization of matrices and tissue engineering is the penetration of the cells into the scaffolds. So far, penetration depth using chondrocytes (depending on the scaffold) is about 200 µm (0.2 mm). This is very poor, regarding a usual defect size and corresponding scaffold size needed. In the present study, the authors did not show the penetration depth of the cells. hondrocytes or MSC “sitting” on the surface of a scaffold and undergoing chondrogenesis is not sufficient for tissue engineering. They explained in the discussion that it was not possible to investigate the ingrowth of cells, because the scaffolds are too brittle to prepare cuttings. However, this is a crucial outcome parameter. How can this issue be circumvented regarding a more detailed characterization and in particular in future in vivo studies?

Moreover, in figure 10, the images demonstrate that the distribution of the cells on the scaffold is very poor. The cells form large aggregates and might undergo chondrogenesis, but how can a cartilage defect be filled with a scaffold, which doesn’t contain equally distributed cells (chondrogenic differentiated cells) but is punctually colonized with clumps of chondrocyte-like cells? Will these cell clumps grow and replace the scaffold in vivo?

5.) Figure 7: How was the vitality calculated? By manual counting? Images should show both channels – red and green cells – not only one channel.

6.) Isn’t there a discrepancy between figure 7 and figure 9? The colonized surface area on BG is about 2 in MSC at all time points; however, there is a strong reduction in DNA content during the course of time. How can that be explained?

Major corrections:

1.) Figure 9: Is there a reason why the DNA content and sGAG content are given in separate graphs? DNA content isn’t that informative as it should be similar to cell vitality and colonized area shown in figure 7. The authors might provide these data in the supplement and should provide a graph with sGAG rel. to DNA (using DNA to normalize the sGAG content).

2.) Statistics: In case of the pH measurement, a two-way ANOVA was performed, which is correct, because there are two parameters (1) time and (2) different scaffolds. However, in the following evaluations, only a one-way ANOVA was used, despite the same experimental setup. The authors need to correct this and perform two-way ANOVA. Same is true for figure 3. In this case the parameters are (1) different scaffolds and (2) different position.

3.) The Authors should show the single data points. In case of the gene expression analysis, figure 11, box plots should be used.

4.) To test potential cytotoxic effects, the authors prepared “extraction solutions” (= conditioned medium?) by culturing scaffolds with growth medium for 48 h. Has the control medium produced equivalently (incubation for 48h without scaffolds)? There are various heat-sensitive components in culture medium. Therefore, the control medium should be produced the same way. The authors should please provide a statement in the manuscript. This study confirmed the biocompatibility/ non-toxicity of the novel scaffolds. The authors should provide data which demonstrate that the “extraction solutions” do not trigger pro-inflammatory response by macrophages/ synovial cells. This is a very important assay regarding biocompatibility and a first characterization of a new biomaterial.

5.) Figure 5: compression curves of the different scaffolds are nice. However, the significance of this analysis could be highly increased when comparing the scaffolds to native cartilage (human or porcine). Eventually, the scaffold will be exposed to the same mechanical stress as the cartilage. Is that technically feasible? Have the authors tried to include such measurements?

Author Response

Dear Editor,                                                                                       Nuremberg, 6th April 2022

The authors would like to thank the reviewer for carefully reading the manuscript and very valuable comments. We modified the manuscript according to the reviewer suggestions with a list of changes shown below. All changes performed are indicated in red in the revised version of the manuscript. We improved the English now. We hope you will find this manuscript suitable for publication in “Cells”. Please do not hesitate to contact me anytime for questions regarding this manuscript.

Sincerely,

Univ.-Prof. Dr. Gundula Schulze-Tanzil

(corresponding author)

Reviewer 2:

In the present manuscript, the authors compared pure bioactive glass (BG) scaffolds and PLGA-infiltrated BG as potential biomaterials in cartilage repair. They demonstrated that the scaffolds are non-cytotoxic and promote chondrogenesis of porcine articular chondrocytes (pAC) or human mesenchymal stem cells (hMSC). Overall, this is a conclusive preliminary study, which provided a general characterization of the scaffolds. In the following, the reviewer would like to ask some questions and make suggestions to further improve the manuscript.

General aspects:

Line 55: Here, the authors refer to a special form of OA, posttraumatic osteoarthritis, this term should be used.

Response (R): The authors thank the reviewer for the precise definition, we have changed it accordingly (line: 55).

Line 59: Could the authors please specify, whether MACI includes both in vivo and in vitro infiltration of cells? To the reviewer’s knowledge, cells are seeded/ encapsulated in vitro, thus cell-bearing matrices are implanted.

R: The authors have clarified this. The reviewer is right, the matrix will be colonized with cells and then implanted. We had the same idea about it, but obviously there was a misunderstanding. We are sorry for that (line 60).

Line 523: “not that high” = lower, reduced?

R: The authors have changed the term in „was lower“. Now line 584.

Line 312: The authors use both names “CellTiter-Blue” and “alamarBlue” assay. It is true that both assays are based on the same principle and component, namely resazurin. However, to the knowledge of the reviewer, these are different brands/ assays. Could the authors please check whether this is correct? Could the authors please explain the difference between CellTiter-Blue (metabolic activity), and the CellTiter96 (cytotoxicity assay; line 510)?

R: Thanks for this hint. „Cell Titer Blue and alamarBlue“ are based on the same principle, yes it is correct. Differences between the two Assays:

  • For the cytotoxcity testing we used the CellTiter96®Aqueous One solution cell prolifertation Assay (MTS). CellTiter 96® AQueous One Solution Cell Proliferation Assay (MTS) (promega.de). The quantity of formazan product as measured by the amount of 490nm absorbance is directly proportional to the number of living cells in culture. Also in another study (Miller et al. 2015) and the International Standard (Microsoft Word - C036406e.doc (nhiso.com)) are using the MTS Assay for the determination of the ISO-10993-5-2009-10 norm certification.
  • For the metabolic assay we used the CellTiter-Blue ® Cell Viability Assay (CellTiter-Blue® Cell Viability Assay (promega.de)). You are right, the assay is based on the ability of living cells to convert a redox dye (resazurin) into a flourescent end product (resorufin). In this assay the fluorescence was measured. Furthermore the Alamar Blue Assay is more sensitve in comparison to the MTS Assay, which is not only our lab experience but was also published from others (Hamid et al. 2004; DOI: 10.1016/j.tiv.2004.03.012) and therefore, we have choosen this assay for the cell and scaffold comparison. We changed the word „AlamarBlue“ into „CellTiter-Blue“ in the manuscript.

Minor correction:

1.) Culture/ growth medium for chondrocytes and MSC contained FCS and amphotericine. Both have impairing effects on chondrogenesis and proliferation of cells. It is very unlikely that cells undergo chondrogenic differentiation in presence of FCS (even 1% of FCS is inhibitory). The authors need to discuss that. Why is there no suppression? Moreover, the authors describe that BG is usually used in bone regeneration. Usually, MSC undergo chondrogenesis and further differentiate into osteoblast lineage. Why didn’t the MSC undergo osteogenic differentiation in this study? In particular in presence of FCS, it is very unlikely that a non-hypertrophic differentiation can take place.

R: The culture medium for hMSC contained platelet lysate (human growth factor-rich human Platelet Lysate (PL)) but no FCS (line 206). In agreement with our study other studies have shown that PL solution can promote chondrogenic differentiation (Hilder et al. 2013). Elsaesser et al. 2016 (DOI 10.1186/s13578-016-0078-6) showed also a chondrogenic differentiation of hMSCs with 10% FCS. Nevertheless, we decided to use the 10% FCS for chondrocyte cultures as we have seen that isolation of cells in this composition works best, allows initial proliferation and cell expansion and therefore, we want to refrain from any further change and keep the same medium composition for the complete cultivation period. We hypothesize that the absence of calicum ions in our scaffolds might provide unfavorable conditions for osteogenic differentiation and therefore, MSCs have the ability to differentiate in the chondrogenic direction as well. We discussed this aspect in lines 908-913.

We fully agree that amphotericin B can inhibit cell growth and differentiation, however, fungal contamination is even more serious and thus should be avoided at all costs, especially when we work with a 3D longterm cultivation. Furthermore, we have to mention that our scaffolds came directly from the scaffold production place (unsterile) and although we sterilized them with HCl, we could not be sure that all fungal spores were killed. AmphoB was used in both cultures at the same concentration and thus both cell types suffer from the same impact. Furthermore, we are at the moment not in preclinical studies and under GMP conditions were AmphoB should be avoided.

2.) The authors explain in the introduction the burdens of MACI, which includes senescence and dedifferentiation of primary cells during in vitro expansion. Therefore, the passage number needs to be provided (maybe the reviewer missed this information) and the dynamics of the de- and re-differentiation would increase the significance of the study. So far, the authors compare the cells at day 7, 21 and 35 to native cartilage. But what was the differentiation status at the time of seeding (after in vitro expansion)? One might assume that there was further dedifferentiation during the earlier time points after seeding on the scaffolds (within the first 7 days), indicated by proliferation. However, at the later timepoints cells might undergo chondrogenesis, as demonstrated by COL2, ACAN etc. expression.

R: You are right, the passage number was missing, we added it in the in Line 219: cells were used in passage 3 and 4.

3.) 10% DMSO was used as a positive control of cell death. Is this a suitable control? Usually, TNF plus cycloheximide or comparable components are used to study cell death. Why have the authors decided for DMSO?

R: The authors have decided for DMSO as a control due to the fact that it is the method recommended by the DIN norm (German institute of norming) for the validation of cytotoxcity of biomaterials.

4.) The main challenge in in vitro colonization of matrices and tissue engineering is the penetration of the cells into the scaffolds. So far, penetration depth using chondrocytes (depending on the scaffold) is about 200 µm (0.2 mm). This is very poor, regarding a usual defect size and corresponding scaffold size needed. In the present study, the authors did not show the penetration depth of the cells. hondrocytes or MSC “sitting” on the surface of a scaffold and undergoing chondrogenesis is not sufficient for tissue engineering. They explained in the discussion that it was not possible to investigate the ingrowth of cells, because the scaffolds are too brittle to prepare cuttings. However, this is a crucial outcome parameter. How can this issue be circumvented regarding a more detailed characterization and in particular in future in vivo studies?

R: The authors thank the reviewer for this very good hint, and we fully agree with you that 200 µm are not so much in view of a cartilage defect in the knee for example, where the cartilage size is higher but in relation to our scaffolds, which have a mean high of 3.32 ± 0.45 mm, we think that a filling with more than 60% is promising. But we believe that with increasing colonization time, cell migration also occurs in the cartilage defect in vivo from the subchondral bone marrow, MSCs are flushed into the scaffold and then, can adhere to the struts and thus fill the space. In addition, we also believe that in initial biocompatibility tests in vivo might prove further immigration of connective tissue cells in inner parts of the scaffold. Our experience from former in vivo studies with comparable brittle scaffolds (TCP, tricalcium phosphate) suggests that subsequent histological processing will be possible, as the entire scaffold itself will then be stabilized from the inside by neoformed tissue and thus, the colonized cells can be traced immunohistologically using antibodies.

Moreover, in figure 10, the images demonstrate that the distribution of the cells on the scaffold is very poor. The cells form large aggregates and might undergo chondrogenesis, but how can a cartilage defect be filled with a scaffold, which doesn’t contain equally distributed cells (chondrogenic differentiated cells) but is punctually colonized with clumps of chondrocyte-like cells? Will these cell clumps grow and replace the scaffold in vivo?

R: The authors thank the reviewer for this comment and we only agree to some degree with the cirticism, considering the fact that the chondrocytes on the BG and 1P variant do adhere and form an evenly distributed extracellular matrix (see life death and SEM images). An exact explanation for the punctate densifications could not be found yet, but also during embryonal cartilage formation (chondrogenesis) in the embryonal mesenchym a cell condensation of precursor cells with subsequent chondrogenic differentiation takes place. We discuss it know in lines 824-827. A statement on whether the cell clumps also remain in vivo cannot be made at this stage, but will be tested in vivo in the future. Nevertheless, other cartilage repair strategies such as chondrospheres (codon) indeed rely on chondrogenic cell clusters. These cells emigrate and distribute in vivo in the cartilage defect area.

5.) Figure 7: How was the vitality calculated? By manual counting? Images should show both channels – red and green cells – not only one channel.

R: The image show indeed both channels, but due to low amount of red dots the green color seems to be dominant. In the attachement you will find one picture with an higher magnification where green and red dots are represented. The calculation was described in the „method part – under 2.10. calculation of the viability“.

 This picture shows hMSCs on the 1P scaffold with a higher magnification. You can see that there are only a few red dots (=dead) but the majority of the cells are vital (=green). We had not decided for this magnification because we want to show an overview of the scaffold surface and not only a small microscopical field. The reader should get an overall impression of the colonized scaffold surface. The calculation was performed with our LAS X 3D software, where you can calculate the green dots and the red dots and therefore, estimate cell vitality. Due to the fact, that it seems sometimes difficult to separate the cells, we used also the measurement according to the vital area and the dead area and came to comparable results. We have additionally generated a Supplemental Figure 1 with a higher magnification of the pictures.

6.) Isn’t there a discrepancy between figure 7 and figure 9? The colonized surface area on BG is about 2 in MSC at all time points; however, there is a strong reduction in DNA content during the course of time. How can that be explained?

R: We thank the reviewer for this comment, which we also recognized, but we would like to note that two different methods were used and therefore, these possible discrepancies occur. Also, it is possible that the standard deviations are also due to the donor number of 3 used. The possible discrepancies could also be explained, by the observation that hMSCs are very enlarged and flattened on the scaffold surface (formation of „cell layer structures“ around the struts and pores like hammocks) and therefore, the calculated surface seemed to be nearly the same (discussed now in lines 876-879). But when we have a look on the precise DNA measurement, we could see differences. This is also a reason why we decided to use the 1P variant in the future, as it was almost constant in the colonized surface and in the DNA content and did not show any reduction.

Major corrections:

1.) Figure 9: Is there a reason why the DNA content and sGAG content are given in separate graphs? DNA content isn’t that informative as it should be similar to cell vitality and colonized area shown in figure 7. The authors might provide these data in the supplement and should provide a graph with sGAG rel. to DNA (using DNA to normalize the sGAG content).

R: As Reviewer 1 and Reviewer 2 recommended, we performed a new calculation, the GAG content per DNA amount shown in Figure 9 now. Moreover, we performed also additional experiments with 1 and 3 days old cultures and added them to the graphic in Figure 9.

2.) Statistics: In case of the pH measurement, a two-way ANOVA was performed, which is correct, because there are two parameters (1) time and (2) different scaffolds. However, in the following evaluations, only a one-way ANOVA was used, despite the same experimental setup. The authors need to correct this and perform two-way ANOVA. Same is true for figure 3. In this case the parameters are (1) different scaffolds and (2) different position.

R: We really want to thank the Reviewer for this very usful information. According to your suggestion we recalculated the statistics in the Figure 3 and indeed, there were more significant differences according to the two way ANOVA testing. We also changed Figure 2 after testing with the two-way ANOVA.

3.) The Authors should show the single data points. In case of the gene expression analysis, figure 11, box plots should be used.

R: We thank the Reviewer for this comment and normally we would agree, but first, the whole manuscript is designed now in bars, second, the boxplot would be more informative if we had a higher sample amount, third, through the inter-donor variance we had a high standard deviation.

4.) To test potential cytotoxic effects, the authors prepared “extraction solutions” (= conditioned medium?) by culturing scaffolds with growth medium for 48 h. Has the control medium produced equivalently (incubation for 48h without scaffolds)? There are various heat-sensitive components in culture medium. Therefore, the control medium should be produced the same way. The authors should please provide a statement in the manuscript. This study confirmed the biocompatibility/ non-toxicity of the novel scaffolds. The authors should provide data which demonstrate that the “extraction solutions” do not trigger pro-inflammatory response by macrophages/ synovial cells. This is a very important assay regarding biocompatibility and a first characterization of a new biomaterial.

R: According to the DIN standard, fresh medium is taken as control for the cyotoxicity assay. Nevertheless, this is a very good indication for our future experiments also in view of the fact that first biocompatibility tests in rodents are in progress. However, as things stand at the moment, there are no macrophages in the laboratory and therefore, they cannot be used. But we have added an experiment and new graphic with the synovial fibroblast cell line K4IM. You can also see that there were no differences between fresh culture medium and culture medium, which was incubated 48 h under similar conditions like in our previous experiments (the same time as the scaffolds). Obviously, the putative loss of heat sensitive components did not have any significant influence on the cell viablity. Based on the reviewers comments we can summarize that the different scaffold variants did not affect the viability of synovial fibroblasts. It is added as Supplemental Figure 2 to the manuscript.

Supplemental Figure 1 Cytotoxicity assay using BG (blue bars), 1P (red bars), 2P (green bars) and 3P (yellow bars) scaffold extracts. Viability of synovial fibroblast cell line K4IM treated for 24 hours with scaffold extracts (48 hours extraction) from three different scaffold charges. Cell viability was above 90% in cells treated with all scaffold extracts . Cell viability of the positive control with 10% dimethylsulfoxide (DMSO, white bar) and 10% DMSO mixed with 48 hours incubated culture medium (white bar with black dots) was below 70% and that of the negative control with culture medium (black) was 100% and the negative control with 48 hours incubated culture medium (black with white dots). The CellTiter 96®Aqueos One Solution Cell Proliferation Assay was used to assess cytotoxicity. n=3. Ordinary one-way ANOVA with multiple comparison, p values: **** < 0.0001.

5.) Figure 5: compression curves of the different scaffolds are nice. However, the significance of this analysis could be highly increased when comparing the scaffolds to native cartilage (human or porcine). Eventually, the scaffold will be exposed to the same mechanical stress as the cartilage. Is that technically feasible? Have the authors tried to include such measurements?

Reviewer 3 Report

General comment

This interesting work aims to tackle a major challenge in regenerative medicine by demonstrating the chondrogenic potential of porcine chondrocyte and human derived stem cells on PLGA infiltrated bioactive glass scaffolds. The utilization of biocative scaffolds can be designed for different clinical approaches such as in vitro pre-biofabrication or post implant in situ regeneration. The study will benefit from an extended description of the targeted clinical application. The methodology is clearly described and the results follow a nice logical flow, but despite the interesting approach to obtain cartilagineus ECM production on bioactive scaffolds, further evidences are required to better support the conclusion that infiltration of PLGA leads to enhancement of chondrogenesis.

Abstract and Introduction

The authour could state what is the nature of the targeted clinical application of the PLGA infiltrated bioactive glass scaffolds to clarify wheter, for example, the intention is to fabricate a cartilagineous implant in vitro prior to implantation in chondral lesions as an alternative to MACI using authologous stem cells and what are the type of lesions that could potentially be treated with this kind of application.

Results

3.1. PLGA distribution within the scaffolds

The authours could show the percentage of PLGA normalized for the percentage of pores for each region to indicate the possible relation. I also assume that Figure 2M Title should read “PLGA distribution per picture”. 

3.3. pH profile during leaching process and gel layer formation

Figure 4 should be centered

3.4. Polymer infiltration strengthens scaffold structure

If possible, the authours could present the differences in stiffness of the samples (before the breaking point) as a compressive modulus to benchmark the results with other studies where bioscaffolds for cartilage repair were analysed. (Loessner et al. Functionalization, preparation and use of cell-laden gelatin methacryloyl–based hydrogels as modular tissue culture platforms. Nat Protoc 11, 727–746, 2016).  The description of the dimensions of the scaffolds in this experimental section can also be helpful for the interpretation of the data obtained.  

3.5. No cytotoxic effects by polymer infiltration

The authours should indicate also in the text that the experiment was performed using growth media in contact with leaching scaffolds to treat pre-seeded cells in a 96 well plate for 24 hours and that the control used for normalization (100%) represent sample replicates treated with control growth media.

3.6. High vitality over 35 days cultivation period

It would be interesting to investigate the efficiency of adhesion of the cells to the scaffolds with different porosity buy evaluating the percentage of non adherent cells respect to the number of cells presented to the scaffold surface (e.g: Dozza et al., When size matters: differences in demineralized bone matrix particles affect collagen structure, mesenchymal stem cell behavior, and osteogenic potential. J Biomed Mater Res A. 2017 Apr;105(4):1019-1033), in this case after the dynamic seeding to better discriminate wheter the data obtained are due to poor adhesion or lack of proliferation. The analysis would also benefit from the inclusion of data of viability at earlier time points (e.g. 1 Day, 3 Days) and from the presentation of the data normalised for the first time point and for the non infiltrated bioactive glass scaffold as a control.

3.8. DNA, sulfated glycosaminoglycan contents and metabolic activity depend on cell type and time point of culturing

The authours may avoid the term “downregulation” of DNA amount and could also present the data using a linear regression fit between the cell number and the DNA concentration for a better correlation with the starting cell number used in the dynamic seeding procedure (another potential indication of adhesion efficency). Moreover, in the pAC group, the GAG content looks higher in the BG scaffolds from day 7 respect to the others, reflecting the difference in DNA amount (which again could be dependent by a different adhesion rate).

In general I suggest to analyse the data also as GAGs/DNA for each group. It would be also interesting to see wheter the scaffold alone shows backgrounds value in the GAG assay and to evaluate GAGs and DNA amount at earlier time points. If my interpretation is correct, the metabolic activity was normalised against an uncolonised scaffold, this should represent the 0% instead of the 100%. I suggest to clarify the interpretation of the metabolic activity data. Of note, the Cell Titer Blue could be read also via fluorescence and the authour should consider wheter the cell titer was also absorbed by the scaffold itself.

 3.9. Cartilage related protein expression

Since the ditribution of the PLGA was non homogeneous through the thickness of the scaffolds, it would be interesting to evaluate if a differential expression of the cartilage related proteins is also present.  

3.10. Non-induced human mesenchymal stromal cells express cartilage-specific genes

Would be interesting to evalute if the infiltration of PLGA can maintain the expression of the chondrogenic markers at later stages (e.g. Day 28-Day 35) since in their previous work (Gögele et al. Highly porous novel chondro-instructive bioactive glass scaffolds tailored for cartilage tissue engineering, Materials Science and Engineering: C, Volume 130, 2021, 112421) the authours observed loss of expression of Collagen II ans SOX9 in the same cell types with bioactive glass scaffolds.

Discussion and Conclusion.

The authors stated that “Significant increases in sGAG synthesis but also in protein expression of collagen type II, cartilage related proteoglycans and the chondrogenic transcription factor SOX9 could be detected”.

Despite the presence of chondrogenic markers were detected by analysis of the scaffolds, the amount of GAGs showed significant increase in hMSCs between day 21 and day 35, but not significant differences between the BG and P groups; a significative increase in SOX 9 could be detected between BG and P1 groups at 7 days for the hMSCs, but they do not reflect the same trends on Collagen II expression (Fig. 11A), which, in contrast, did not show any significant enahncement from the infiltration of PLGA. This is also reflected by the immunostaining analysis where “No significant differences could be evaluated between the different scaffold variants colonized with hMSC (Fig. 10 J)”.  

Author Response

Dear Editor,                                                                                       Nuremberg, 6th April 2022

The authors would like to thank the reviewer for carefully reading the manuscript and very valuable comments. We modified the manuscript according to the reviewer suggestions with a list of changes shown below. All changes performed are indicated in red in the revised version of the manuscript. We improved the English now. We hope you will find this manuscript suitable for publication in “Cells”. Please do not hesitate to contact me anytime for questions regarding this manuscript.

Sincerely,

Univ.-Prof. Dr. Gundula Schulze-Tanzil

(corresponding author)

Reviewer 3:

This interesting work aims to tackle a major challenge in regenerative medicine by demonstrating the chondrogenic potential of porcine chondrocyte and human derived stem cells on PLGA infiltrated bioactive glass scaffolds. The utilization of biocative scaffolds can be designed for different clinical approaches such as in vitro pre-biofabrication or post implant in situ regeneration.

The study will benefit from an extended description of the targeted clinical application.

R: Thanks for pointing that out. We shortly explained it in the conclusion section in lines 959-962.

The methodology is clearly described and the results follow a nice logical flow, but despite the interesting approach to obtain cartilagineus ECM production on bioactive scaffolds, further evidences are required to better support the conclusion that infiltration of PLGA leads to enhancement of chondrogenesis.

R: We agree with the reviwerr`s criticism, deeper investigation of signaling pathways, possibly affected by PLGA are needed. However, considering all aspects of analyses there is now consistent evidence for enhanced chondrogenesis by PLGA. It was also not expected since the rationale for investigating PLGA infiltration was to exclude significant inhibitory effects.

Abstract and Introduction

The authour could state what is the nature of the targeted clinical application of the PLGA infiltrated bioactive glass scaffolds to clarify wheter, for example, the intention is to fabricate a cartilagineous implant in vitro prior to implantation in chondral lesions as an alternative to MACI using authologous stem cells and what are the type of lesions that could potentially be treated with this kind of application.

R: Our idea could be the use of in vitro colonized scaffolds, but this would mean that we would have a considerable bureaucratic effort with all approvals and authorities, since it is then a medical device with personalized cells. From a purely economic perspective, the implantation of a pure bioglass scaffold in the defect would therefore make sense. The time and effort required for approval may be less and could be realized more quickly. In this case, it would be expected that the stem cells mobilized from the bone tissue could be flushed into the scaffold by creating access to the subchondral bone marrow cavity and thus adhere to the struts and differentiate. MSCs could also be harvested during surgery from iliac crest within the same surgical procedure and implanted simultaneously with the scaffolds. Based on our current data, therefore, we cannot say exactly which approach we can aim for in the future. In general, it is tailored for chondral defects, as a composite also for osteochondral defects.

We discuss this now in lines: 959-962.

Results

3.1. PLGA distribution within the scaffolds

The authours could show the percentage of PLGA normalized for the percentage of pores for each region to indicate the possible relation. I also assume that Figure 2M Title should read “PLGA distribution per picture”. 

R: We would like to sincerely thank the reviewer for discovering the error and offer our sincere apologies. The image has been updated with the correct caption and the new significancies according to Reviewer 1 suggestion. 

3.3. pH profile during leaching process and gel layer formation

Figure 4 should be centered

R: That was true, we fully aggree with you that the Fig. 4 should be centered. We corrected it in our Word file and hope the change will be seen also in the PDF version.

3.4. Polymer infiltration strengthens scaffold structure

If possible, the authours could present the differences in stiffness of the samples (before the breaking point) as a compressive modulus to benchmark the results with other studies where bioscaffolds for cartilage repair were analysed. (Loessner et al. Functionalization, preparation and use of cell-laden gelatin methacryloyl–based hydrogels as modular tissue culture platforms. Nat Protoc 11, 727–746, 2016).  The description of the dimensions of the scaffolds in this experimental section can also be helpful for the interpretation of the data obtained.  

3.5. No cytotoxic effects by polymer infiltration

The authours should indicate also in the text that the experiment was performed using growth media in contact with leaching scaffolds to treat pre-seeded cells in a 96 well plate for 24 hours and that the control used for normalization (100%) represents sample replicates treated with control growth media.

R: The authors have restructured their text in Line 543 and 545.

3.6. High vitality over 35 days cultivation period

It would be interesting to investigate the efficiency of adhesion of the cells to the scaffolds with different porosity buy evaluating the percentage of non adherent cells respect to the number of cells presented to the scaffold surface (e.g: Dozza et al., When size matters: differences in demineralized bone matrix particles affect collagen structure, mesenchymal stem cell behavior, and osteogenic potential. J Biomed Mater Res A. 2017 Apr;105(4):1019-1033), in this case after the dynamic seeding to better discriminate wheter the data obtained are due to poor adhesion or lack of proliferation. The analysis would also benefit from the inclusion of data of viability at earlier time points (e.g. 1 Day, 3 Days) and from the presentation of the data normalised for the first time point and for the non infiltrated bioactive glass scaffold as a control.

R: According to the reviewer suggestion we performed cell seeding efficiency (%) measurements after 1 and 3 days. Dozza et al., is cited now (line 288). No significant differences could be evaluated between the different scaffolds, but in the pAC cultures we found a slightly reduction of the cell seeding efficiency in the 1P, 2P and 3P scaffold group after 1 and 3 days. The cell seeding efficiency was not evaluated after later timepoints like 7 or 21 days because the medium has to be changed and therefore, it does not make sense. This image was added to figure 7.

3.8. DNA, sulfated glycosaminoglycan contents and metabolic activity depend on cell type and time point of culturing

The authours may avoid the term “downregulation” of DNA amount

R: This term has been removed.

and could also present the data using a linear regression fit between the cell number and the DNA concentration for a better correlation with the starting cell number used in the dynamic seeding procedure (another potential indication of adhesion efficency).

R: since we show now GAG/DNA and separately seeding efficacy we decided to do not calculate the linear regression fit.

Moreover, in the pAC group, the GAG content looks higher in the BG scaffolds from day 7 respect to the others, reflecting the difference in DNA amount (which again could be dependent by a different adhesion rate).

In general I suggest to analyse the data also as GAGs/DNA for each group.

R: Done: As Reviewer 1 and 2 recommended we changed figure 9 into GAG content per DNA amount.

It would be also interesting to see wheter the scaffold alone shows backgrounds value in the GAG assay

R: There is no absorbance by the scaffolds due to the fact, that after the selected incubation time points, the liquids were pipetted into a new plate and only the liquid was measured and not the colonized scaffolds + AlamarBlue solution.

and to evaluate GAGs and DNA amount at earlier time points.

If my interpretation is correct, the metabolic activity was normalised against an uncolonised scaffold, this should represent the 0% instead of the 100%. I suggest to clarify the interpretation of the metabolic activity data. Of note, the Cell Titer Blue could be read also via fluorescence and the authour should consider wheter the cell titer was also absorbed by the scaffold itself.

R: The metabolic activity of a non colonized scaffold is still 100% percent, because the pure Alamar blue dye is 0% in the calculation.

 3.9. Cartilage related protein expression

Since the ditribution of the PLGA was non homogeneous through the thickness of the scaffolds, it would be interesting to evaluate if a differential expression of the cartilage related proteins is also present.  

R: We thank the reviewer for this comment and we tried also to visualize the cartilage extracellular matrix protein aggrecan, but we found out that our antibody did not allow a clear evaluation. Therefore, we decided not to show the data (seen in Figure 1 and 2 after 14 and 28 days). We had also stained collagen type I, and it was also not expressed neither by pACs nor by the hMSCs. For example, in Figure 3 is the staining of collagen type II (green) and collagen type I (red) and there was no expression after 14 days of cultivation.

Fig. 1: pAC on the BG scaffold. In blue (DAPI) are shown the cell nuclei, in green is collagen type II shown and in red is aggrecan shown after 14 days.

Fig. 2: pAC on the BG scaffold. The cell nuclei are shown in blue (DAPI), collagen type II is shown in green and aggrecan is shown in red after 12 days.

Fig. 3: hMSCs on the BG scaffold. The cell nuclei are shown in blue (DAPI). Collagen type II is shown in green and collagen type I is shown in red after 14 days.

3.10. Non-induced human mesenchymal stromal cells express cartilage-specific genes

Would be interesting to evalute if the infiltration of PLGA can maintain the expression of the chondrogenic markers at later stages (e.g. Day 28-Day 35) since in their previous work (Gögele et al. Highly porous novel chondro-instructive bioactive glass scaffolds tailored for cartilage tissue engineering, Materials Science and Engineering: C, Volume 130, 2021, 112421) the authours observed loss of expression of Collagen II ans SOX9 in the same cell types with bioactive glass scaffolds.

R: We fully agree with the reviewer that it would be interesting to have a deeper look in at a later time point, but it is not feasable at the moment to perform the missing time points.

Discussion and Conclusion.

The authors stated that “Significant increases in sGAG synthesis but also in protein expression of collagen type II, cartilage related proteoglycans and the chondrogenic transcription factor SOX9 could be detected”.

Despite the presence of chondrogenic markers were detected by analysis of the scaffolds, the amount of GAGs showed significant increase in hMSCs between day 21 and day 35, but not significant differences between the BG and P groups; a significative increase in SOX 9 could be detected between BG and P1 groups at 7 days for the hMSCs, but they do not reflect the same trends on Collagen II expression (Fig. 11A), which, in contrast, did not show any significant enahncement from the infiltration of PLGA. This is also reflected by the immunostaining analysis where “No significant differences could be evaluated between the different scaffold variants colonized with hMSC (Fig. 10 J)”.  

R: The GAG to DNA ratio has been described now with no significant differences. The gene expression data show significant differences compared to native cartilage the differences between BG and 1P represented trends, which were indeed divergent between SOX9 and collagen type II. There might be a different time dependency which we could not visualize since we investigated only two time points of analysis.

Round 2

Reviewer 1 Report

Peer-Review Cells – 1639117 (2nd round of revisions)

The manuscript entitled “Biodegradable Poly(D-L-lactide-co-glycolide) (PLGA) infiltrated bioactive glass (CAR12N) scaffolds enhances mesenchymal stromal cell chondrogenesis for cartilage tissue engineering” by Clemens Leo Gögele et al. was considerably improved by the authors after this round of revisions. Almost all the reviewers’ comments/suggestions were properly addressed/clarified.

Few minor issues:

  1. At the depth they were performed without providing any values for compressive modulus, compressive strength, etc, the current state of the results of the mechanical tests do not add any advantage to the manuscript and may be removed.
  2. Figure 1 caption formatting is still different from the others. Please standardize.
  3. (line 456, subsection 3.1): It is better to use “within” instead of “with”. Please correct.

Author Response

Dear Editor,                                                                                      Nuremberg, 1rst May 2022

The authors would like to thank once again the reviewer for carefully reading the manuscript and very valuable comments. We modified the manuscript according to the reviewer suggestions with a list of changes shown below. All changes performed are indicated in red in the revised version of the manuscript. We improved the English now. We hope you will find this manuscript suitable for publication in “Cells”. Please do not hesitate to contact me anytime for questions regarding this manuscript.

Sincerely,

Univ.-Prof. Dr. Gundula Schulze-Tanzil

(corresponding author)

Reviewer 1

The manuscript entitled “Biodegradable Poly(D-L-lactide-co-glycolide) (PLGA) infiltrated bioactive glass (CAR12N) scaffolds enhances mesenchymal stromal cell chondrogenesis for cartilage tissue engineering” by Clemens Leo Gögele et al. was considerably improved by the authors after this round of revisions. Almost all the reviewers’ comments/suggestions were properly addressed/clarified.

Few minor issues:

  1. At the depth they were performed without providing any values for compressive modulus, compressive strength, etc, the current state of the results of the mechanical tests do not add any advantage to the manuscript and may be removed.

Response: The authors follow the reviewer's concerns. The biomechanical data described, present initial preliminary force measurements. With the consideration not to overstate these data we have decided to show the graph as a supplemental figure (1) to give the interested reader a first insight that PLGA infiltration is very important to increase stability. Further, we underlined the preliminary character of biomechanical investigation in the conclusion section.

  1. Figure 1 caption formatting is still different from the others. Please standardize.

Response: We would like to thank the reviewer for his suggestion for correction and have also taken this into account and made the image descriptions consistent.

  1. (line 456, subsection 3.1): It is better to use “within” instead of “with”. Please correct.

       Response: we corrected it (line 416).

Reviewer 2 Report

The authors thoroughly corrected and reworked the manuscript, which has been significantly improved. There are only two additional aspects, which need to be addressed:

1.) The authors did not respond to the last point:
Figure 5: compression curves of the different scaffolds are nice. However, the significance of this analysis could be highly increased when comparing the scaffolds to native cartilage (human or porcine). Eventually, the scaffold will be exposed to the same mechanical stress as the cartilage. Is that technically feasible? Have the authors tried to include such measurements?

2.) The authors largely corrected the statistical analysis; however, the statistics in figure 7 and 9 also require a two-way ANOVA.

Author Response

Dear Editor,                                                                                      Nuremberg, 1rst May 2022

The authors would like to thank once again the reviewer for carefully reading the manuscript and very valuable comments. We modified the manuscript according to the reviewer suggestions with a list of changes shown below. All changes performed are indicated in red in the revised version of the manuscript. We improved the English now. We hope you will find this manuscript suitable for publication in “Cells”. Please do not hesitate to contact me anytime for questions regarding this manuscript.

Sincerely,

Univ.-Prof. Dr. Gundula Schulze-Tanzil

(corresponding author)

Reviewer 2

The authors thoroughly corrected and reworked the manuscript, which has been significantly improved. There are only two additional aspects, which need to be addressed:

1.) The authors did not respond to the last point:
Figure 5: compression curves of the different scaffolds are nice. However, the significance of this analysis could be highly increased when comparing the scaffolds to native cartilage (human or porcine). Eventually, the scaffold will be exposed to the same mechanical stress as the cartilage. Is that technically feasible? Have the authors tried to include such measurements?

R: We agree with the reviewer and thank him/her for this comments. We also think it would we very important in future to compare the compressability of native cartilage with that of our scaffolds to see if they are indeed comparable stable. To underline this we inserted in the conclusion section a clear hint on this important fact (lines 929-930) as a limitation of the present study. At the moment the technical device (GT200 particle compression tester) it not usable and we cannot perform additional measurement due to the fact that some spare parts can not be delivered. It should also be mentioned at this point that the device (GT200 particle compression tester) used was designed for other materials, hence the evaluation range is limited. Hence, we are very sorry that we can not meet your expectation in this regard. For this reason and in agreement with the criticism of reviewer 1 we transferred the biomechanical data in the supplementary part of this manuscript and clearly stated the preliminary character of the biomechanical analysis with an outlook on the more detailed future measurements required.

2.) The authors largely corrected the statistical analysis; however, the statistics in figure 7 and 9 also require a two-way ANOVA.

R: We thank the reviewer and corrected it.

Reviewer 3 Report

The manuscript was subjected to extended revision and inclusion of additional controls.

Despite the evident improvement, there are still some concerns regarding the enhancement of chondrogenesis due to the infiltration of PLGA. I really appreciated that the authours’ performed additional immunostainings to touch this point, but in my opinion, based on the data obtained, the effect of PLGA infiltration on the GAG/DNA amount (Fig.9), on the accumulation of Collagen Type II in the extracellular matrix (Fig.10) and on the expression of SOX9 and COL2A1 (Fig.11) is not evident when compared to the non infiltrated standard bioglass. However, this study is very relevant for cartilage regeneration applications since the infiltration shows a significant impact on the stability of the scaffold and therefore it could be a crucial addendum for a future in vivo application. In fact, strenghten a bioscaffold with reinforcement materials can result in the impairment of the biologicval activity, which is not happening with the infiltration methodology established by the authors. I suggest to focus the manuscript, including the title, on the stability of the implant and the “maintenment” of rather then the “enhancement” of chondrogenesis in vitro. I would also underlyine further the capability of the bioactive glass to trigger the chondrogenesis without the addition of chondrogenic factors, a key characteristic for the application of chondrogenic scaffolds in vivo.

Finally, the definition of mesenchymal cells is under continuos debate. However, guidelines are emerging suggesting that if a mesenchymal cell can express stem markers at the isolation and possess demonstrable progenitor cell functionality of self-renewal and differentiation, than it should be defined as stem, while the stromal attribute should refer to a bulk population with notable secretory, immunomodulatory and homing properties as  suggested by the International Society for Cell & Gene Therapy (Viswanathan S, Shi Y, Galipeau J, Krampera M, Leblanc K, Martin I, Nolta J, Phinney DG, Sensebe L. Mesenchymal stem versus stromal cells: International Society for Cell & Gene Therapy (ISCT®) Mesenchymal Stromal Cell committee position statement on nomenclature. Cytotherapy. 2019 Oct;21(10):1019-1024. doi: 10.1016/j.jcyt.2019.08.002. Epub 2019 Sep 13. PMID: 31526643). Given the above, the authours could consider to define the mesenchymal cells used as “mesenchymal stem cells (MSCs)”

Author Response

Dear Editor,                                                                                      Nuremberg, 1rst May 2022

The authors would like to thank once again the reviewer for carefully reading the manuscript and very valuable comments. We modified the manuscript according to the reviewer suggestions with a list of changes shown below. All changes performed are indicated in red in the revised version of the manuscript. We improved the English now. We hope you will find this manuscript suitable for publication in “Cells”. Please do not hesitate to contact me anytime for questions regarding this manuscript.

Sincerely,

Univ.-Prof. Dr. Gundula Schulze-Tanzil

(corresponding author)

Reviewer 3

The manuscript was subjected to extended revision and inclusion of additional controls.

Despite the evident improvement, there are still some concerns regarding the enhancement of chondrogenesis due to the infiltration of PLGA. I really appreciated that the authours’ performed additional immunostainings to touch this point, but in my opinion, based on the data obtained, the effect of PLGA infiltration on the GAG/DNA amount (Fig.9), on the accumulation of Collagen Type II in the extracellular matrix (Fig.10) and on the expression of SOX9 and COL2A1 (Fig.11) is not evident when compared to the non infiltrated standard bioglass.

R: we agree that there a no significant differences at the protein level detectable.

However, this study is very relevant for cartilage regeneration applications since the infiltration shows a significant impact on the stability of the scaffold and therefore it could be a crucial addendum for a future in vivo application. In fact, strenghten a bioscaffold with reinforcement materials can result in the impairment of the biologicval activity, which is not happening with the infiltration methodology established by the authors.

I suggest to focus the manuscript, including the title, on the stability of the implant and the “maintenment” of rather then the “enhancement” of chondrogenesis in vitro. I would also underlyine further the capability of the bioactive glass to trigger the chondrogenesis without the addition of chondrogenic factors, a key characteristic for the application of chondrogenic scaffolds in vivo.

R: we follow the criticism of the reviewer and changed the title of the manuscript accordingly. Once again, we would like to thank the reviewer for taking a very detailed look at this manuscript and seeing a possible potential in our scaffolds for in vivo use.  We also agree with the reviewer that PLGA infiltration did not lead to any significant expression changes.

But it is important to note that the content of PLGA, especially in the variants which was only once infiltrated is indeed very low, so we did not expect huge differences. It is true that stability can definitely be increased by PLGA infiltration. Nevertheless, we would like to mention again at this point that this is not yet a finished "product". In the future, we would like to focus on a more even distribution of PLGA within the scaffold and try to further increase stability with the lowest PLGA content possible.

Finally, the definition of mesenchymal cells is under continuos debate. However, guidelines are emerging suggesting that if a mesenchymal cell can express stem markers at the isolation and possess demonstrable progenitor cell functionality of self-renewal and differentiation, than it should be defined as stem, while the stromal attribute should refer to a bulk population with notable secretory, immunomodulatory and homing properties as  suggested by the International Society for Cell & Gene Therapy (Viswanathan S, Shi Y, Galipeau J, Krampera M, Leblanc K, Martin I, Nolta J, Phinney DG, Sensebe L. Mesenchymal stem versus stromal cells: International Society for Cell & Gene Therapy (ISCT®) Mesenchymal Stromal Cell committee position statement on nomenclature. Cytotherapy. 2019 Oct;21(10):1019-1024. doi: 10.1016/j.jcyt.2019.08.002. Epub 2019 Sep 13. PMID: 31526643). Given the above, the authours could consider to define the mesenchymal cells used as “mesenchymal stem cells (MSCs)”

R: We thank the reviewer for this very important information and correct it in our manuscript.
